# Omega 3 Blends of Sunflower and Flaxseed Oil—Modeling Chemical Quality and Sensory Acceptability

**DOI:** 10.3390/foods13233722

**Published:** 2024-11-21

**Authors:** Ranko Romanić, Tanja Lužaić, Lato Pezo, Bojana Radić, Snežana Kravić

**Affiliations:** 1Faculty of Technology Novi Sad, University of Novi Sad, Bulevar cara Lazara 1, 21000 Novi Sad, Serbia; tanja.luzaic@tf.uns.ac.rs (T.L.); bojana.radic@fins.uns.ac.rs (B.R.); sne@uns.ac.rs (S.K.); 2Institute of General and Physical Chemistry, University of Belgrade, 11000 Belgrade, Serbia; latopezo@yahoo.co.uk; 3Institute of Food Technology in Novi Sad, University of Novi Sad, Bulevar cara Lazara 1, 21000 Novi Sad, Serbia

**Keywords:** refined sunflower oil, cold-pressed flaxseed oil, oil blend, optimization, artificial neural network

## Abstract

Oil blending is increasingly utilized to improve and model the characteristics of enriched oils. This study aims to investigate the effect of blending refined sunflower oil (rich in essential omega 6 fatty acids) with cold-pressed flaxseed oil (a source of essential omega 3 fatty acids) on the fatty acid composition, quality, color, and sensory characteristics of the resulting oils. Principal component analysis (PCA) showed that the optimal fatty acid composition was achieved in the sample with 20% sunflower oil and 80% flaxseed oil (20S/80F). However, developing a new product is highly complex due to the importance of oil quality and sensory characteristics. Therefore, an Artificial Neural Network (ANN) was applied to optimize the proportions of flaxseed and sunflower oil to create an oil blend with improved nutritional, oxidative, and sensory characteristics compared to the individual oils. The ANN analysis determined the optimal composition of the oil blend to be 51.5% refined sunflower oil and 48.5% cold-pressed flaxseed oil. Sensory characteristics pose a particular challenge in optimization, as flaxseed oil, which increases essential omega 3 fatty acids, has a specific taste that is not widely favored by consumers. Nonetheless, by blending with refined sunflower oil, the resulting optimal blend (51.5% refined sunflower oil and 48.5% cold-pressed flaxseed oil) possesses pleasant sensory characteristics.

## 1. Introduction

Vegetable oil is an essential component of the diet and represents the primary source of lipids, substances that not only provide energy but also contribute to the construction of lipid membranes in the body [1,2]. It is a significant source of fatty acids, especially essential ones, and also facilitates the absorption of liposoluble vitamins. In human nutrition, vegetable oils are most commonly used for cooking and are extensively utilized in the food industry, where their composition and quality in terms of nutritional and sensory characteristics define the quality of the final food product [3]. Currently, no single vegetable oil can be characterized by ideal functional, nutritional, and sensory characteristics along with adequate oxidative stability [4]. The nutritional quality and stability of vegetable oils can be improved through hydrogenation, interesterification, fractionation, and blending. Most of these processes come with certain limitations, including the need for expensive equipment and substantial financial investments, while hydrogenation has the drawback of forming harmful trans isomers. Conversely, blending is an efficient method for producing a functional product—a blended oil with a balanced fatty acid content and positive health effects [4,5]. Blending vegetable oils with different properties is one of the very simple ways to create new specific products with desired sensory, physicochemical, and oxidative properties [6]. Recent literature on blended oils with balanced fatty acid composition is predominantly limited to Asian countries, where blended oils based on rapeseed oil, palm oil, and flaxseed oil, as well as sunflower and rice bran oils dominate [7,8,9,10,11,12], i.e., oils from oilseeds characteristic for specific geographic areas.

In many industries, finished products are enriched to achieve new food characteristics, physical or chemical properties, or specific health benefits [13,14,15,16]. This practice is also applied in the edible oil industry for producing blended oils [4]. Lehnert et al., 2019 [17], blended peanut and flaxseed oil to create new oil with a balanced ratio of omega 3 and omega 6 fatty acids, as well as a balance of monounsaturated and polyunsaturated fatty acids. Kotliar et al., 2018 [18], produced 10 binary and ternary oil blends of sunflower, flaxseed, camelina, and pumpkin oil in various mass ratios to achieve new products with a diverse fatty acid composition and physicochemical parameters. These blends also had varying concentrations of β-carotene and tocopherol. The main goal was to develop new oils tailored for special nutritional requirements.

Oil blending affects the triacylglycerol profile, leading to changes in the physical properties of the oil such as turbidity point, solid fat content, sensory quality, smoke point, density, and viscosity [19,20]. Most consumers consider sensory quality to be more important than nutritional value. The sensory characteristics of the final oil product are primarily influenced by the blending process. The sensory quality of food is affected by chemical reactions during oil frying, such as oxidation, hydrolysis, thermal decomposition, and isomerization [4,21]. Recognizing the properties of the oil and selecting appropriate blending oils can mitigate most of these undesirable reactions, thereby improving the nutritional and sensory quality of the final product. Oil blending can also alter the odor profile. Sensory acceptability tests on different oil blends have shown that blending can enhance the properties of each oil and result in a more desirable product [4].

Determining the ω6/ω3 fatty acid ratio is significant from a health perspective, and studying the ideal ratio is the subject of many scientific papers. Retrospectively, this ratio has increased multiple times from 1 to over 20 due to a diet rich in red meat, dairy products, and salt, while being low in fruits, vegetables, legumes, and fish [22]. Depending on dietary habits, traditions, and food availability, the intake of essential fatty acids varies worldwide [23]. Stark et al., 2016 [23], and Schuchardt et al., 2024 [24], created a map of essential omega 3 fatty acid intake globally, finding that countries like Japan, South Korea, some Scandinavian countries (Norway and Finland), and territories such as Greenland (Denmark) and Alaska (USA) have exceptionally high intakes of these fatty acids. In contrast, countries like Brazil, Egypt, Iran, and India have very low intakes. European countries, the USA, Canada, China, and Australia have moderate intakes. Given the diversity in the intake of essential fatty acids, recommendations vary by geographic area. However, since most populations face an omega 3 fatty acid deficiency, the World Health Organization (WHO) has recommended a balanced intake of essential omega 6 (linoleic acid) and omega 3 (alpha-linolenic acid) in a ratio between 5:1 and 10:1 [25]. Sunflower oil is rich in linoleic acid, while flaxseed oil is rich in alpha-linolenic acid. Therefore, refined sunflower oil and cold-pressed flaxseed oil were used in the production of oil blends in this study. This study aims to investigate the overall acceptability of the new oil blends, considering their quality characteristics, color characteristics, and sensory acceptability. Additionally, the fatty acid composition of the oil blends was investigated. The main goal of this paper was to examine the effect of adding refined sunflower and cold-pressed flaxseed oil on the quality of the oil blend, in terms of fatty acid content, oil quality, color, and sensory characteristics. The performance of Artificial Neural Networks (ANNs) was compared to experimental results.

## 2. Materials and Methods

### 2.1. Sample Preparation

This study aimed to investigate samples of blended vegetable oils. Refined sunflower oil (100S/0F), cold-pressed flaxseed oil (0S/100F), and a control sample were purchased from the local supermarkets. Examined oil blends were composed of refined sunflower oil (S) and cold-pressed flaxseed oil (F), obtained by these two oils blending in set proportions. Examined binary blends contained the following refined sunflower oil and cold-pressed flaxseed oil content: 90S/10F, 80S/20F, 70S/30F, 60S/40F, 50S/50F, 40S/60F, 30S/70F, 20S/80F, 10S/90F. The ratio of shares of refined sunflower oil (S) and cold-pressed flaxseed oil (F) called S and F ratio was calculated. Detailed preparation, packaging, and storage of oil blends were described in a previous study by Romanić et al., 2021 [26]. Oil blends were prepared in the quantities allowed for all the mentioned tests. As a control sample, the only commercially available oil blend on the Serbian market was used. This oil blend was composed of three refined vegetable oils: rapeseed, sunflower, and corn oil, in decreasing order. The oils share in the blend is not known, but the oil producer declared the presence of balanced essential omega 3/omega 6 fatty acids content.

### 2.2. Determination of Fatty Acid Composition

The fatty acid composition of blended oils was determined using gas chromatography–mass spectrometry (GC–MS) (according to ISO 12966-4, 2015 [27]) by application of the GC7890B gas chromatograph (Agilent Technologies, Santa Clara, CA, USA) with MSD5977A (Agilent Technologies, Santa Clara, CA, USA) single quadrupole mass spectrometer. Prepared fatty acid methyl esters (FAME) were separated using capillary column SP–2560 (100 m × 0.25 mm × df 0.20) according to ISO 12966-2, 2017 [28]. Detection and quantification conditions were described by Romanić et al., 2021 [26].

The omega 6 and omega 3 (ω6/ω3) fatty acid ratio and nutritional indices, which assessed the nutritional quality of the analyzed oils, were determined based on the fatty acid composition, i.e., the content of individual fatty acids. The calculation for the indices of atherogenicity (IA) and thrombogenicity (IT) were developed by Ulbricht and Southgate, 1991 [29], while the hypocholesterolaemic/hypercholesterolaemic (HH) ratio was proposed by Santos-Silva et al., 2002 [30]. The indices were calculated based on the following Equations (1)–(3):(1)$$IA=\frac{C12:0+4\times C14:0+C16:0}{\Sigma MUFA+\Sigma \omega 6+\Sigma \omega 3}$$
(2)$$IT=\frac{C14:0+C16:0+C18:0}{0.5\times \Sigma MUFA+0.5\times \Sigma \omega 6+3\times \Sigma \omega 3}$$
(3)$$HH=\frac{C18:1cis9+C18:2\omega 6+C20:4\omega 6+C18:3\omega 3+C20:5\omega 3+C22:5\omega 3+C22:6\omega 3}{C14:0+C16:0}$$

### 2.3. Oil Quality and Stability Investigation

All quality and stability parameters were investigated according to adequate ISO standards. 

#### 2.3.1. Acid Value

The acid value (AV) was determined according to ISO 660, 2020 [31].

#### 2.3.2. Peroxide Value

The peroxide value (PV) was determined according to ISO 3960, 2017 [32].

#### 2.3.3. Anisidine Value 

The anisidine value (AnV) was determined according to ISO 6885, 2016 [33].

#### 2.3.4. Total Oxidation Index

The total oxidation index (TOTOX) was computed according to Equation (4) based on PV and AnV [34,35,36].
TOTOX = 2 × PV + p-AnV(4)

#### 2.3.5. Conjugated Dienes and Conjugated Trienes

Conjugated dienes (CD) and conjugated trienes (CT) content was determined according to ISO 3656:2013/Amd 1:2017 [37].

### 2.4. Accelerated Stability Tests

In order to investigate the oxidative stability of the samples, two accelerated stability tests were performed: Rancimat and RapidOxy test.

#### 2.4.1. Rancimat Test

Rancimat accelerated stability test was carried out on Rancimat apparatus, model 743 (Metrohm, Herisau, Switzerland), to measure the induction period of oil samples. Measurements were performed according to ISO 6886, 2016 [38]. The procedure and measurement conditions were previously described by Lužaić et al., 2022 [39]. Namely, 2.50 ± 0.01 g of oil sample was oxidized at a temperature of 100 °C and the airflow of 18–20 L/h. Volatile products formed during the oxidation were diluted in 0.05 L of distilled water. The device measures the conductivity recorded by apparatus software and is used to obtain the induction period (IP) expressed in hours with an accuracy of 0.1.

#### 2.4.2. RapidOxy Test

The oxidative stability of the oil was also tested using RapidOxy 100 (Anton Paar, Ostfildern, Germany). Namely, 3.00 ± 0.01 g of the sample was weighed into a glass container and placed in the device chamber. Oxygen was introduced into the chamber until the pressure rose to 700 kPa. Subsequently, the chamber with the sample was heated to 140 °C. As the temperature increased, the pressure in the chamber also increased (up to about 1000 kPa) until oxidation began. As a result of the reaction of oxygen with the tested oil sample, oxygen was consumed, and the pressure inside the chamber decreased. When the pressure in the chamber dropped by 10%, the oxidation reaction was considered complete, and the time was recorded as an induction period in minutes [40].

### 2.5. Color Measurement

CIELab color measurements (lightness L*, redness a*, and yellowness b*) were carried out by MINOLTA Chroma Meter CR-400 (Minolta Co., Ltd., Osaka, Japan, illuminant D-65, 8 mm diameter aperture, 2° standard observer). Blended oils were tempered at room temperature (25 °C) for 2 h before the color measurement. Total color differences (ΔE) of oils with respect to the control sample (0) were calculated based on Equation (5):(5)$$∆E=\sqrt[{2}]{{\left({L*}_{0}-L*\right)}^{2}+{\left({a*}_{0}-a*\right)}^{2}+{\left({b*}_{0}-b*\right)}^{2}}$$

### 2.6. Sensory Analysis

The trained panel consisted of 20 sensory panelists, and staff members of the Department of Food Engineering—Faculty of Technology Novi Sad, Serbia, between 25 and 65 years old. With each trained panelist, 8–10 tasters, potential consumers of younger, middle, and older age, participated in the evaluation. The total number of panelists and tasters was 215. Evaluation of the sensory quality of oils was performed by ranking tests [41]. The system of analytical–descriptive tests was used with points ranging from 0 (unacceptable quality) to 5 (optimal quality) [42]. The following characteristics were examined: color, odor, and taste. Twelve samples of blended vegetable oils were served to the sensory panelists (including the control sample). Samples of blended vegetable oil for sensory analysis were prepared in such a way that oils were pre-heated to a temperature of 35–40 °C, where the odor and taste of the oil are more pronounced, in a quantity of (20–25 mL) in a laboratory glass of 50 mL volume [42]. Each data point from the sensory analysis represented the average of all panelists’ opinions. Average rating represented the average score of sensory attributes of color, odor, and taste, as the three most characteristic sensory attributes of vegetable oils, while total acceptability represented the weighted points of sensory attributes for color, odor, and taste according to Equation (6):(6)$$Total\;acceptability=\frac{(Color\times 0.6+Odor\times 0.8+Taste\times 1)\times 5}{12}$$

### 2.7. Statistical Analysis

All results were presented as a mean value ± standard deviation (n = 3). One-way analysis of variance (ANOVA) with a post hoc Tukey’s HSD test was used to establish significant differences among the data at the significance level *p* < 0.05.

Principal component analysis (PCA) is a statistical analysis technique, mainly used to reduce the initial amount of data and/or to obtain orthogonal variables, especially if collinearity is present. This analysis provides insight into the existence of regularity in a large amount of data by providing information regarding components that behave in a similar way.

#### 2.7.1. Artificial Neural Network Modeling

This article evaluated a Multilayer Perceptron Model (MLP) consisting of three layers (input, hidden, and output). The main reason for the application of these architectures in the paper was the proven ability to approximate nonlinear functions [43].

The experimental database for ANN was randomly split into training, cross-validation, and testing data sets (representing 60%, 20%, and 20%, respectively). Recommendations indicate the use of only one hidden layer because the use of multiple layers leads to the local minimum problem [44,45]. The Levenberg–Marquardt algorithm was used for the network training due to its proven high accuracy in similar function approximation [28]. A different number of neurons in the hidden layer were examined and the sum of squares error (SOS) values for the networks were calculated to find the optimal ANN [26]. The optimal number of hidden neurons was chosen based on the minimal difference between predicted ANN values and desired outputs. The neural network was performed using StatSoft Statistica 10.0 [46]. ANN calculation could be presented with the following Equation (7) [47]:(7)$$Y=f_{1}\left(W_{2}\cdot f_{2}\left(W_{1}\cdot X+B_{1}\right)+B_{2}\right)$$
where W_1_ and B_1_ and W_2_ and B_2_ are matrices where coefficients associated with the hidden and the output layers (weights and biases, respectively) are grouped. Y is the matrix of the output variables, X is the matrix of input variables, and f_1_ and f_2_ are transfer functions in the hidden and output layers, respectively.

Weights and biases were obtained during the training step. In this step, an update of these values occurs in order of optimization by minimizing the error function between network and experimental outputs, according to the SOS values and BFGS algorithm. This algorithm was used to accelerate and stabilize convergence [43].

#### 2.7.2. Training, Testing, and System Implementation

The training step was started after the neural network was defined. The main goal was to obtain a high predictive ANN based on the high variable parameters, so the training step had to be repeated several times. It is considered that the training process should be carried out when learning and cross-validation curves (SOS vs. training cycles) approach zero.

#### 2.7.3. Global Sensitivity Analysis

The relative influence of inputs was obtained using Yoon’s interpretation method [48]. The weight coefficients obtained by ANN were used in this method.

#### 2.7.4. The Accuracy of the ANN Model

Validation of the developed ANN model was performed using a coefficient of determination (*r*^2^), reduced chi-square (χ^2^), mean bias error (MBE), root mean square error (RMSE), and mean percentage error (MPE). The calculation of these parameters is shown in Equations (8)–(11) [49]:(8)$$\chi^{2}=\frac{\sum_{i=1}^{N}{\left(x_{exp,i}-x_{pre,i}\right)}^{2}}{N-n}$$
(9)$$RMSE={\left[\frac{1}{N}\cdot \sum_{i=1}^{N}{\left(x_{pre,i}-x_{exp,i}\right)}^{2}\right]}^{1/2}$$
(10)$$MBE=\frac{1}{N}\cdot \sum_{i=1}^{N}\left(x_{pre,i}-x_{exp,i}\right)$$
(11)$$MPE=\frac{100}{N}\cdot \sum_{i=1}^{N}\left(\frac{\left|x_{pre,i}-x_{exp,i}\right|}{x_{exp,i}}\right)$$
where x_exp,i_ indicates experimental values, x_pre,i_ values predicted by the developed model, and N and n indicate a number of observations and constants, respectively.

## 3. Results and Discussion

### 3.1. Fatty Acid Composition

Sunflower oil is a source of essential omega 6 fatty acids (linoleic acid), while flaxseed oil contains significant amounts of the essential omega 3 fatty acids (alpha-linolenic acid), as mentioned previously. Thus, the highest content of alpha-linolenic fatty acids of 54.84 ± 0.08% was found in pure flaxseed oil, while with the decrease in the flaxseed oil share in the oil blend, the content of this fatty acid significantly decreased (*p* ˂ 0.05) to only 0.07 ± 0.01%, noticed in the pure sunflower oil, Table 1. The reverse case was observed with linoleic fatty acid. Namely, the highest content of 59.71 ± 0.02% was noticed in pure sunflower oil. By reducing the share of sunflower oil in vegetable oil blends, the content of linoleic fatty acid was significantly reduced (*p* ˂ 0.05) to 18.17 ± 0.02% (0S/100F). Similar conclusions were reached by Hashempour-Baltork et al., 2018 [50], by blending olive, flaxseed, and sesame oils in different proportions. Flaxseed oil was used as a source of essential omega 3 fatty acids, while sesame oil was a source of essential omega 6 fatty acids. With the increase in the proportion of flaxseed oil, the content of omega 3 fatty acids increased, while with the increase in the proportion of sesame oil, the content of omega 6 fatty acids increased, and vice versa. Regarding polyunsaturated fatty acids, the presence of gamma-linolenic acid in minor percentages (<0.15%) was also detected, while it was not detected in samples with less than 60% of flaxseed oil. The oleic fatty acid content ranged from 17.02 ± 0.02% (0S/100F) to 30.25 ± 0.05% (100S/0F). Other monounsaturated fatty acids, palmitoleic (C16:1) and gadoleic acid (C20:1), were found in minor percentages (<0.2%). Concerning the saturated fatty acids, palmitic acid was dominant with a content of approximately 5–6.5%, followed by stearic acid at 2.5–4.5%. Arachidic, behenic, and lignoceric acids were detected in all the samples in a very small amount, while myristic acid appeared only in the pure flaxseed oil. The fatty acid composition of samples 100S/0F and 0S/100F was following the Codex Alimentarius Standards, 1999 [51], for sunflower and flaxseed oils.

Investigating the ω6/ω3 fatty acid ratio is significant from a health perspective, and studying the ideal ratio is the subject of many scientific papers. Retrospectively, this ratio has increased multiple times from 1 to over 20 due to a diet rich in red meat, dairy products, and salt, while being low in fruits, vegetables, legumes, and fish [22]. Different ratios are important in the prevention of certain diseases. For example, a ratio of 4 is associated with reduced mortality from cardiovascular diseases, a ratio of 2–3 is significant for patients with colorectal tumors and rheumatoid arthritis, and a ratio of 5 has a beneficial effect for patients suffering from asthma [22,52]. To achieve a balanced ω6/ω3 fatty acid ratio, lower ratios should be aimed for [53,54], and it is considered that a value range of 1 to 5 is optimal for human health [52]. Among the samples analyzed, the extremely high value (892.91) is characteristic of pure refined sunflower oil, while even a minimal addition of flaxseed oil significantly reduces the ratio (to 13.19). Optimal values of this ratio (1–5) were achieved by adding 30 to 60% cold-pressed flaxseed oil. In the control sample, a ratio of 4.96 was determined, which is close to the maximum recommended ratio, while the addition of 20% flaxseed oil resulted in a ratio slightly higher than the recommended (5.57).

To assess the nutritional quality of mixed oils, nutritional indices including IA, IT, and HH were calculated. The index of atherogenicity (IA) is based on the ratio of saturated to unsaturated fatty acids describes the atherogenic potential of fatty acids. Consuming foods with lower IA can contribute to reducing total and LDL cholesterol levels [55]. The index of thrombogenicity (IT) characterizes the thrombogenic potential of fatty acids and represents the relationship between pro-thrombogenic (C14:0, C16:0, and C18:0) and anti-thrombogenic fatty acids (monounsaturated, ω3, and ω6 fatty acids) [29]. Essentially, both indices are associated with cardiovascular disease risk and must be as low as possible.

The obtained IA and IT values in all oil samples are far below 1. It should be noted that with the change in fatty acid composition resulting from the addition of flaxseed oil, IA values vary slightly (0.06–0.07), while IT values decrease (0.19–0.05). The lowest IA value was obtained for the control sample at 0.04. Considering the hypocholesterolemic/hypercholesterolemic fatty acid ratio [30], the HH index considers the effect of fatty acids on cholesterol metabolism, and from a nutritional standpoint, high values of this index are desirable. Blending sunflower oil with flaxseed oil increases the HH index in the analyzed samples from 14.96 to 18.03. The highest value of this index (23.26) was determined in the control sample.

### 3.2. Quality and Stability Parameters of Oil

AV, PV, AnV, TOTOX, CD, and CT, as well as CD/CT of the examined oil samples, are summarized in Table 2.

#### 3.2.1. AV

The acid value (AV) indicates the content of free fatty acids. Refined oils have a lower AV due to the neutralization process [56]. Consequently, the lowest AV (0.18 ± 0.00 mgKOH/g) was observed in refined sunflower oil. Blended oils exhibited significantly (*p* ˂ 0.05) higher AV, while the AV determined in cold-pressed flaxseed oil was significantly higher than all other examined samples (1.26 ± 0.06 mgKOH/g). Similar AVs in refined sunflower oils were reported by Pal et al., 2014 [56] (0.48 mgKOH/g), and Javidipour et al., 2017 [57] (0.20 mgKOH/g). According to Raczyk et al., 2016 [58], the AV in cold-pressed flaxseed oil ranged from 0.51 ± 0.03 mgKOH/g to 3.20 ± 0.01 mgKOH/g. Grover et al., 2021 [59], reported similar values blending flaxseed and sunflower oil in various proportions (80:20, 70:30, 60:40, 50:50, 40:60, 70:30, 80:20), determining AVs ranging from 1.07 ± 0.10 to 1.12 ± 0.10 mgKOH/g. The acid values obtained in all investigated samples are in accordance with the Codex Alimentarius Standards, 1999 [51].

#### 3.2.2. PV, AnV, and TOTOX

Peroxide value indicates primary oxidation products content, anisidine value secondary oxidation products, while the total oxidation index gives an overall insight into oxidation products. No statistically significant difference (*p* ˂ 0.05) in PV was found between pure flaxseed oil and oil blends with 90 and 80% of flaxseed oil. With a decrease in flaxseed oil share of 10 or 20%, PV increased significantly. Pure sunflower oil had the highest PV of 1.60 ± 0.01 mmol/kg, probably due to the low refining conditions. In the same sample also was noticed the highest AnV of 15.12 ± 0.18, which is a consequence of the already mentioned reason. Flaxseed oil has a very good initial quality (low values of PV, AnV, etc.), which indicates that the cold-pressed oil is produced from high-quality raw materials. Such high AnV was a consequence of the refining process where secondary oxidation products were being formed. Reduction in refined sunflower oil share of 10 or 20% leads to a significant decrease (*p* ˂ 0.05) in the AnV of the examined samples. The same legality was established for TOTOX. The obtained results were in line with previous studies [56,58] and Codex Alimentarius Standards, 1999 [51], for PV.

#### 3.2.3. CD and CT Content

Conjugated dienes and conjugated trienes indicate products obtained by the oxidation of fatty acids. CD and CT content of the pure cold-pressed flaxseed oil and refined sunflower oil was in line with previous studies [60,61] and differ significantly (*p* ˂ 0.05). The addition of sunflower oil (10 to 90%) led to a significant change in CD and CT content. CD and CT content are correlated with PV and AnV values, considering that PV indicates the presence of primary oxidation products in connection with conjugated dienes, while AnV indicates the presence of secondary oxidation products in connection with the content of conjugated trines in investigated blended oils.

### 3.3. Accelerated Stability Tests

The oxidative stability of the oils was investigated using accelerated thermal stability tests. Similar results were obtained using the Rancimat and RapidOxy tests, as shown in Table 3. Refined sunflower oil (100S/0F) displayed good oxidative stability characteristics with an induction period of 9.48 ± 0.14 h obtained by the Rancimat test, or 34.37 ± 0.46 min by the RapidOxy test. These results are consistent with previous research reported by Velasco et al., 2004 [62], and Lužaić et al., 2022 [39]. With addition of cold-pressed flaxseed oil (0S/100F), the induction period was significantly reduced to 4.28 ± 0.08 h (Rancimat test) or 18.39 ± 0.36 min (RapidOxy test), as determined in the cold-pressed flaxseed oil. Similar values were previously reported by Tańska et al., 2016 [63], and Mikołajczak and Tańska, 2022 [64]. Significantly higher induction period values were observed in the control sample (16.47 ± 0.23 h, and 70.50 ± 0.84 min), which is a consequence of the different composition of fatty acids (essential and non-essential), the content and composition of minor components with pro-oxidative and antioxidant effects, the production process, raw materials, etc. [65,66]. Oxidative changes occur at the unsaturated bonds of fatty acids, so the influence of the polyunsaturated fatty acid (PUFA) content and the induction period obtained by accelerated stability tests was examined, revealing an extremely strong negative correlation between the PUFA content (based on the results, Table 1) and the induction period as determined by the Rancimat test (R = −0.984, *p* = 0.000) and the RapidOxy test (R = −0.998, *p* = 0.000) in the investigated samples of blended vegetable oils.

### 3.4. Color Measurement

The color parameters, L* (lightness), a* (redness), b* (yellowness), and ΔE (total color difference), of oil blends were given in Table 4. The increase in flaxseed oil share in the oil blends led to significant (*p* ˂ 0.05) changes in redness, yellowness, and lightness. The color of the oil blends is derived from the pigments present mostly in the cold-pressed flaxseed oil. Thus, the main source of redness and yellowness was pigments from flaxseed oil, since refined sunflower oil had low pigment quantities due to the bleaching process [39]. The highest lightness value (26.50 ± 0.01) was noticed in pure refined sunflower oil, while a slightly lower value (25.48 ± 0.02) was obtained in the control sample. These high lightness values were also a consequence of the bleaching process in the refined oil production [67].

### 3.5. Sensory Analysis

The sensory attributes of edible vegetable oils significantly affect their overall quality and are always noticed first by consumers. All sensory attributes, except for the taste of samples ranging from 0S/100F to 50S/50F, were satisfactory (≥2.8), as shown in Table 5. Cold-pressed flaxseed oil is characterized by a specific taste, which is not particularly desirable to consumers. A bitter taste develops after just one day of storage, due to the deterioration of flaxseed oil and the methionine oxidation of its cyclolinopeptides [68]. In fact, the undesirable bitter taste in cold-pressed flaxseed oil is due to cyclolinopeptides E [68,69]. This bitter taste is also transferred to other products containing flaxseed oil. For instance, Bialasová et al., 2018 [70], determined the negative impact of adding flaxseed oil and flaxseed meal to yogurt on its taste and flavor. In the examined oil blends, the inclusion of up to 50% flaxseed oil affected the acceptability of the product’s taste. No significant difference (*p* ˂ 0.05) in the sensory attributes of odor was found between the investigated samples. The sensory attribute of color for the 100S/0F sample was significantly different compared to the 90S/10F and 80S/20F samples, while no significant difference was noted between other samples. The sensory attribute of color had an insignificant correlation (R = 0.760, *p* = 0.05) with the instrumental parameter of yellowness (*b** value). There was no statistically significant difference between samples in terms of average rating and overall acceptability, as shown in Table 5.

### 3.6. Principal Component Analysis

The principal component analysis (PCA) of sensory analysis, oil quality, and color characteristics data (11 examined variables) showed that the first two components take into account 80.31% of the total variance. The obtained PCA map shows that the quality parameters—AnV (9.2% of the total variance, based on correlations), TOTOX (9.9%), and PV (11.1%)—have a positive influence on the first PC, whereas AV (25.3%) exhibited a negative score value according to the PC1 (Figure 1a). The positive influence toward the PC1 coordinate was obtained for sensory variables: average rating (9.3%), taste (10.0%), and total acceptability (9.6%), whilst the negative influence on PC1 calculation was observed for the color coordinate (11.4%).

The positive effect on the second PC was observed for the oil quality parameters CD and CT (13.4% and 9.0% of the total variance, based on correlations, respectively), the color coordinates b* (20.2%), and ΔE (19.2%) and also the sensory color (11.8%). The color coordinate L* (12.5%) negatively influenced the calculation of the PC2 coordinate.

Based on the PCA sample 100S/0F was the closest to the control sample, according to the sensory analysis, oil quality, and color characteristics data. The similarity between these two products in the oil quality, color characteristics, and sensory attributes stemmed from the same production process, namely both samples were refined oils. Thus, there was no statistically significant difference found between sample 100S/0F and the control sample in all sensory attributes. Moreover, the lowest total color difference (1.28 ± 0.02) was noticed between these two oils.

The principal component analysis (PCA) of fatty acid composition data explained that the first two PCs take into account 90.14% of the total variance (Figure 1b). The obtained PCA map indicates that the content of C18:0 (9.2% of the total variance, based on correlations), C18:3n3 (9.1%), and C18:3n6 (9.9%) were positively influential for PC1 calculation, while the content of C20:1 (7.2%), C16:0 (9.4%), C24:0 (7.2%), C20:0 (7.2%), C18:1 (10.0%), C18:2n6 (9.8%), and C22:0 (10.0%) was negatively influential for PC1.

The influence of the content of C16:1 (18.4%) and C20:1 (15.5%) was positive for PC2 calculation, while the influence of C14:0 content was negative for PC2 calculation (50.7%). PCA showed that sample 80S/20F was the closest to the control sample, according to the fatty acid composition. The results of PCA, as well as one-way ANOVA applied to post hoc Tukey’s HSD test, showed no statistically significant difference between these two samples (*p* ˂ 0.05), which indicates a balanced ratio of essential fatty acids in the 80S/20F sample.

### 3.7. ANN Model

According to ANN (sum of *r*^2^ and SOSs for all variables in one ANN), the optimal number of neurons in the hidden layer was six (network MLP 2-6-28), when high values of *r*^2^ (0.959, 0.750, and 0.964 for training, testing, and validation performances, respectively) and low SOS values were obtained, Table 6.

The high nonlinearity of the developed system made the ANN model complex (214 weights and biases) [71,72].

Appendix A present the elements of matrix *W*_1_ and vector *B*_1_ (presented in the bias column), and elements of matrix *W*_2_ and vector *B*_2_ (bias) for the hidden layer, respectively, used for calculation in Equation (7).

The order of magnitude of experimental errors for β-carotene content was the same as the SOS obtained using the ANN model [47,73].

Appendix A shows the residual analysis of the developed model used for the evaluation of the model fitting. The developed ANN model had an insignificant lack of fit tests, meaning that the sensory analysis parameters, fatty acids content, oil quality, and color characteristics predicted using the developed model are satisfactory. High *r*^2^ values indicate that the variation was accounted for and that the data fitted the proposed model satisfactorily [72,74].

Residuals were also analyzed through the mean and the standard deviation. The mean of the observed parameter prediction was between −1.141 and 0.507, while the standard deviations were between 0.002 and 2.060. These results showed a good approximation to a normal distribution around zero with a probability of 95% (2 × SD), which means a good generalization ability of the ANN model for the range of observed experimental values [47].

Optimization of the composition of the oil blend, composed of refined sunflower oil and cold-pressed flaxseed oil, was performed using the developed ANN model (Equation (7), Appendix A), by minimizing the differences in the observed parameters (sensory analysis parameters, fatty acids content, oil quality, and color characteristics) with the control sample, to ensure that the optimum sample would be most similar to the control sample. During the optimization of the oil composition, the importance of parameter groups was applied, and weight coefficients were assigned to the group of parameters: 0.20 to the sensory analysis group of parameters, 0.15 to the color group, 0.40 to the oil quality group, and 0.25 to the fatty acids group of parameters. The limiting factors of optimization were sensory attribute scores, as some blends had an unacceptable taste. The values of the control parameters used in the optimization are given as follows. The value of the control parameters used as limiting factors in the optimization is a score of ≥2.8 for color, odor, and taste, as well as a score of ≥2.8 for average rating and total acceptability. 

The optimum composition of the oil blend was 51.5% refined sunflower seed oil and 48.5% cold-pressed flaxseed oil. Optimal sensory attributes of color and odor according to ANN analysis were 3.9 and 4.0. The optimal sensory attribute of taste was at the lower limit of acceptability (2.8) due to the high share (48.5%) of flaxseed oil with an unpleasant taste. The average rating and total acceptability of the new oil blend amounted to 3.5 and 3.4. The new oil blend had the following optimal color characteristics: L* = 24.434; a* = −0.454; b* = 10.820; ΔE = 6.991. Oxidative quality and stability parameters were given an advantage during optimization (importance factor 0.4) since these parameters represented toxic products formed during oxidation. Optimal oxidation product content amounted to 0.954 mmol/kg (PV), 8.039 (AnV), 9.370 (TOTOX), 3.625 (CD), 0.505 (CT), and 6.538 (CD/CT), while AV amounted to 0.568 mgKOH/g. The ratio of omega 6 (linoleic = 40.327% and gamma-linolenic = 0.006%) and omega 3 (alpha-linolenic = 27.562%) fatty acids of optimal oil blend (1.5:1) was different compared to WHO recommendations (5–10:1) and the control sample (1:1.7). The amount of omega 3 fatty acids present in the new blend is still significant considering that the omega 6 and omega 3 fatty acid ratio in the diet of the population in this area is 20 or even 30:1. Most abundant monounsaturated acid in the new oil blend was oleic acid (22.848%), then palmitoleic (0.075%) and gadoleic (0.104%). Considering the saturated fatty acids, palmitic (5.661%) and stearic (3.695%) acids were dominant, followed by behenic (0.324%), arachidic (0.158%), lignoceric (0.141%), and myristic (0.001%) acid.

### 3.8. Sensitivity Analysis

In this section, the influence of S and F ratio input variables, identified on the quality of oil blend, regarding sensory analysis parameters, fatty acids content, oil quality, and color characteristics, was studied (see Appendix A). Bleaching and deodorization processes in the production of refined sunflower oils contributed to the positive effect of the S ratio on the lightness and sensory attributes of color and taste, as well as on average rating and total acceptability of oil. The same processes also affected the negative contribution of the S ratio to yellowness and redness of oil, as well as sensory attributes of odor [67]. Positive contribution on alpha-linolenic acid content had an F ratio, while the positive influence on linoleic acid content had an S ratio, as a main source of these fatty acids. Since alpha-linolenic fatty acid compared to linoleic acid is more unstable, a negative contribution of the content of the F ratio to oxidation product content (PV, AnV, and TOTOX) was expected.

## 4. Conclusions

PCA applied in this study proved that sample 100S/0F was the closest to the control sample, according to the sensory analysis, oil quality, and color characteristics data, while according to the fatty acid composition, it was the sample 80S/20F.

Furthermore, ANN optimization of the composition of the oil blend gave the optimum content of refined sunflower oil (51.5%) and cold-pressed flaxseed oil (48.5%) regarding the control sample. The new oil blend had optimal fatty acid composition (C14:0 = 0.001%; C16:0 = 5.661%; C16:1 = 0.075%; C18:0 = 3.695%; C18:1 = 22.848%; C18:2n6 = 40.327%; C18:3n6 = 0.0057%; C18:3n3 = 27.562%; C20:0 = 0.158%; C20:1 = 0.104%; C22:0 = 0.324%; C24:0 = 0.141%), optimal oil quality and stability parameters (AV = 0.568 mgKOH/g; PV = 0.954 mmol/kg; AnV = 8.039; TOTOX = 9.370; CD = 3.625; CT = 0.505; CD/CT = 6.538), optimal color characteristics (L* = 24.434; a* = −0.454; b* = 10.820; ΔE = 6.991), and optimal sensory characteristics (color = 3.9; odor = 4.0; taste = 2.8). 

The findings from this study indicate the potential to enhance refined sunflower oil with omega 3 fatty acids by incorporating cold-pressed flaxseed oil. The study demonstrated that the overall acceptability of the oil blends—considering quality characteristics, color attributes, and sensory acceptability—is satisfactory. Furthermore, the fatty acid profile of the blends offers the necessary nutritional value. Artificial Neural Networks (ANNs) provide the capability to determine the optimal proportion of refined sunflower oil (S) and cold-pressed flaxseed oil (F) at any given time, based on various modeled quality parameters.

## Figures and Tables

**Figure 1 foods-13-03722-f001:**
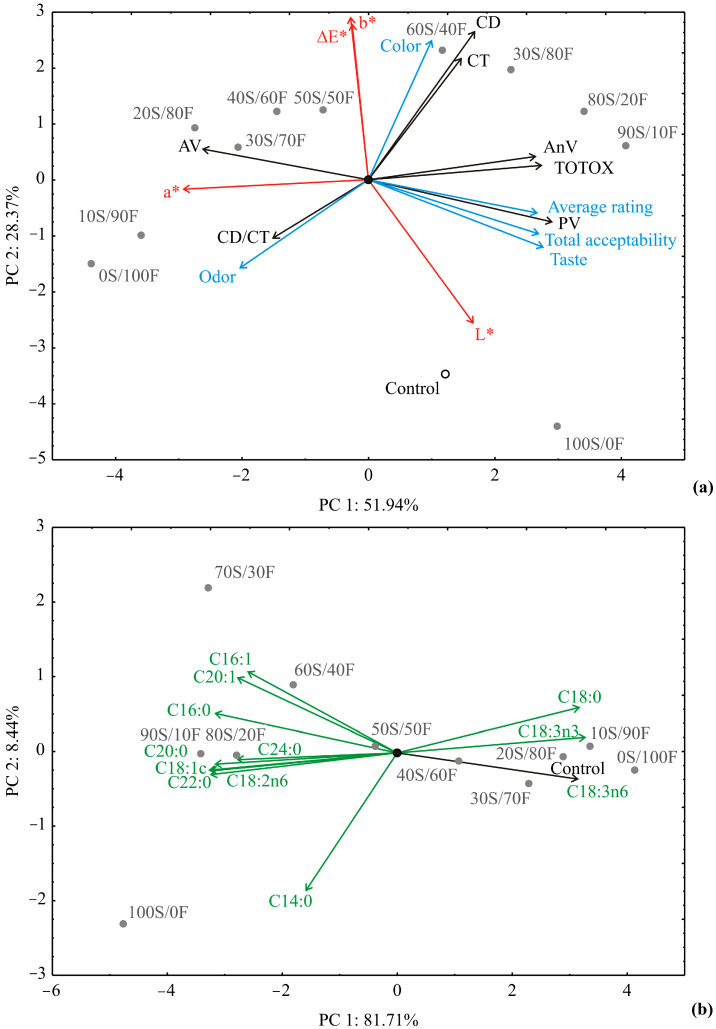
Principal component analysis of blended vegetable oils based on (**a**) sensory analysis (color, odor, taste, average rating, total acceptability), oil quality (acid value—AV, peroxide value—PV, anisidine value—AnV, total oxidation value—TOTOX, conjugated dienes content—CD, conjugated trienes content—CT, conjugated dienes–conjugated trienes ratio—CD/CT), and color characteristics data (lightness—L*, redness—a*, yellowness—b*, total color difference—ΔE); (**b**) fatty acid composition of the samples.

**Table 1 foods-13-03722-t001:** Fatty acid composition of the examined oil samples. Sample names indicate the share of refined sunflower oil/cold pressed flaxseed oil in the oil blend.

Oil Blend	Fatty Acid Composition (%)											Nutrient Indices			
	C16:0	C16:1	C18:0	C18:1	C18:2n6	C18:3n6	C18:3n3	C20:0	C20:1	C22:0	C24:0	ω6/ω3	AI	TI	HH
100S/0F	5.97 ± 0.00 ^h^	0.08 ± 0.01 ^d^	2.76 ± 0.00 ^b^	30.25 ± 0.05 ^j^	59.71 ± 0.02 ^l^	nd	0.07 ± 0.01 ^a^	0.19 ± 0.00 ^f^	0.11 ± 0.00 ^bc^	0.60 ± 0.03 ^g^	0.21 ± 0.01 ^fgh^	892.91	0.07	0.19	14.96
90S/10F	5.84 ± 0.01 ^g^	0.07 ± 0.00 ^d^	2.93 ± 0.01 ^c^	28.57 ± 0.04 ^i^	57.18 ± 0.06 ^k^	nd	4.33 ± 0.02 ^b^	0.18 ± 0.01 ^ef^	0.11 ± 0.01 ^ab^	0.56 ± 0.01 ^g^	0.22 ± 0.01 ^gh^	13.19	0.06	0.16	15.43
80S/20F	5.84 ± 0.01 ^g^	0.07 ± 0.00 ^d^	3.20 ± 0.00 ^d^	27.36 ± 0.02 ^h^	52.99 ± 0.04 ^j^	nd	9.51 ± 0.02 ^c^	0.18 ± 0.00 ^ef^	0.10 ± 0.00 ^ab^	0.51 ± 0.01 ^f^	0.23 ± 0.01 ^h^	5.57	0.06	0.13	15.39
70S/30F	6.08 ± 0.01 ^i^	0.11 ± 0.01 ^e^	3.40 ± 0.01 ^e^	26.62 ± 0.06 ^g^	48.34 ± 0.01 ^i^	nd	14.48 ± 0.02 ^d^	0.17 ± 0.01 ^de^	0.13 ± 0.01 ^c^	0.47 ± 0.02 ^ef^	0.19 ± 0.00 ^ef^	3.34	0.07	0.12	14.70
60S/40F	6.00 ± 0.02 ^h^	0.08 ± 0.01 ^d^	3.62 ± 0.01 ^f^	26.13 ± 0.03 ^g^	42.16 ± 0.02 ^h^	nd	21.11 ± 0.01 ^e^	0.18 ± 0.00 ^de^	0.11 ± 0.00 ^ab^	0.44 ± 0.02 ^e^	0.17 ± 0.00 ^de^	2.00	0.07	0.10	14.90
50S/50F	5.56 ± 0.01 ^f^	0.06 ± 0.01 ^bc^	3.61 ± 0.00 ^f^	23.82 ± 0.02 ^f^	40.13 ± 0.01 ^g^	nd	26.02 ± 0.01 ^f^	0.17 ± 0.00 ^d^	0.10 ± 0.00 ^ab^	0.37 ± 0.01 ^d^	0.16 ± 0.01 ^de^	1.54	0.06	0.08	16.17
40S/60F	5.46 ± 0.01 ^e^	0.06 ± 0.01 ^c^	3.64 ± 0.00 ^f^	22.37 ± 0.02 ^e^	36.02 ± 0.01 ^f^	0.08 ± 0.00 ^a^	31.69 ± 0.01 ^g^	0.15 ± 0.01 ^c^	0.10 ± 0.00 ^ab^	0.30 ± 0.01 ^c^	0.13 ± 0.01 ^ab^	1.14	0.06	0.07	16.51
30S/70F	5.28 ± 0.01 ^d^	0.05 ± 0.01 ^b^	3.83 ± 0.01 ^g^	21.17 ± 0.03 ^d^	32.04 ± 0.02 ^e^	0.09 ± 0.00 ^a^	36.93 ± 0.05 ^h^	0.14 ± 0.01 ^bc^	0.09 ± 0.01 ^a^	0.27 ± 0.01 ^c^	0.11 ± 0.01 ^a^	0.87	0.06	0.07	17.09
20S/80F	5.27 ± 0.01 ^d^	0.06 ± 0.01 ^bc^	4.00 ± 0.02 ^h^	19.38 ± 0.58 ^c^	26.54 ± 0.13 ^d^	0.11 ± 0.01 ^b^	44.09 ± 0.78 ^i^	0.13 ± 0.01 ^ab^	0.09 ± 0.01 ^a^	0.21 ± 0.01 ^b^	0.13 ± 0.00 ^abc^	0.60	0.06	0.06	17.07
10S/90F	5.18 ± 0.03 ^c^	0.06 ± 0.01 ^c^	4.11 ± 0.01 ^i^	18.55 ± 5.78 ^b^	23.56 ± 0.01 ^b^	0.12 ± 0.01 ^c^	47.88 ± 0.05 ^j^	0.12 ± 0.01 ^a^	0.09 ± 0.01 ^a^	0.18 ± 0.01 ^ab^	0.14 ± 0.01 ^bc^	0.49	0.06	0.06	17.37
0S/100F	5.00 ± 0.01 ^b^	0.04 ± 0.01 ^a^	4.28 ± 0.03 ^j^	17.02 ± 0.02 ^a^	18.17 ± 0.02 ^a^	0.14 ± 0.00 ^d^	54.84 ± 0.08 ^k^	0.12 ± 0.01 ^a^	0.09 ± 0.00 ^a^	0.15 ± 0.00 ^a^	0.16 ± 0.01 ^cd^	0.33	0.06	0.05	18.03
Control	3.99 ± 0.00 ^a^	0.11 ± 0.00 ^e^	1.53 ± 0.00 ^a^	64.21 ± 0.01 ^k^	23.70 ± 0.01 ^c^	nd	4.78 ± 0.70 ^b^	0.40 ± 0.01 ^g^	0.79 ± 0.01 ^d^	0.31 ± 0.02 ^c^	0.20 ± 0.01 ^fg^	4.96	0.04	0.09	23.26

Values are means ± standard deviation (n = 3); nd—not detected. Different lower-case letters in the same column indicate significantly different values (*p* < 0.05), according to post hoc Tukey’s HSD test.

**Table 2 foods-13-03722-t002:** Acid value (AV), peroxide value (PV), anisidine value (AnV), total oxidation index (TOTOX), conjugated dienes (CD) and conjugated trienes (CT) content, and conjugated dienes–conjugated trienes ratio (CD/CT) of the blended vegetable oils.

Oil Blend	AV (mgKOH/g)	PV (mmol/kg)	AnV	TOTOX	CD	CT	CD/CT
100S/0F	0.18 ± 0.00 ^d^	1.60 ± 0.01 ^g^	15.12 ± 0.18 ^k^	18.32 ± 0.19 ^j^	2.39 ± 0.06 ^e^	0.25 ± 0.01 ^f^	9.44 ± 0.32 ^b^
90S/10F	0.34 ± 0.00 ^b^	1.51 ± 0.05 ^d^	13.22 ± 0.08 ^j^	16.24 ± 0.03 ^i^	3.70 ± 0.02 ^a^	0.38 ± 0.01 ^a^	9.82 ± 0.16 ^b^
80S/20F	0.39 ± 0.01 ^b^	1.46 ± 0.00 ^d^	12.19 ± 0.19 ^i^	15.11 ± 0.19 ^h^	3.66 ± 0.10 ^a^	0.41 ± 0.02 ^ab^	9.01 ± 0.59 ^bc^
70S/30F	0.49 ± 0.03 ^c^	1.19 ± 0.02 ^c^	11.62 ± 0.08 ^h^	14.00 ± 0.09 ^g^	4.24 ± 0.02 ^b^	0.62 ± 0.01 ^c^	6.81 ± 0.10 ^a^
60S/40F	0.52 ± 0.01 ^c^	1.20 ± 0.02 ^c^	10.16 ± 0.01 ^g^	12.56 ± 0.04 ^f^	4.39 ± 0.08 ^b^	0.61 ± 0.01 ^c^	7.15 ± 0.07 ^ad^
50S/50F	0.67 ± 0.03 ^a^	1.01 ± 0.05 ^b^	8.27 ± 0.01 ^f^	10.28 ± 0.10 ^e^	3.61 ± 0.03 ^a^	0.42 ± 0.02 ^b^	8.60 ± 0.35 ^bcd^
40S/60F	0.69 ± 0.04 ^a^	0.99 ± 0.01 ^b^	6.32 ± 0.04 ^e^	8.30 ± 0.06 ^d^	3.33 ± 0.12 ^i^	0.50 ± 0.02 ^h^	6.70 ± 0.05 ^a^
30S/70F	0.62 ± 0.01 ^a^	0.77 ± 0.01 ^e^	4.61 ± 0.03 ^a^	6.14 ± 0.02 ^a^	3.08 ± 0.01 ^h^	0.34 ± 0.01 ^g^	9.07 ± 0.25 ^bc^
20S/80F	0.69 ± 0.03 ^a^	0.66 ± 0.01 ^a^	4.82 ± 0.03 ^a^	6.13 ± 0.03 ^a^	2.85 ± 0.05 ^g^	0.38 ± 0.01 ^a^	7.57 ± 0.24 ^acd^
10S/90F	0.84 ± 0.06 ^e^	0.64 ± 0.01 ^a^	4.25 ± 0.02 ^d^	5.52 ± 0.02 ^c^	2.20 ± 0.01 ^d^	0.17 ± 0.01 ^e^	13.22 ± 0.86 ^e^
0S/100F	1.26 ± 0.06 ^f^	0.67 ± 0.01 ^a^	0.00 ± 0.00 ^b^	1.33 ± 0.02 ^b^	2.00 ± 0.01 ^c^	0.11 ± 0.01 ^d^	18.31 ± 1.67 ^f^
Control	0.32 ± 0.01 ^b^	1.37 ± 0.02 ^f^	3.42 ± 0.02 ^c^	6.16 ± 0.05 ^a^	2.60 ± 0.01 ^f^	0.40 ± 0.01 ^ab^	6.56 ± 0.09 ^a^

Values are means ± standard deviation (n = 3). Different lower-case letters in the same column indicate significantly different values (*p* < 0.05), according to post hoc Tukey’s HSD test.

**Table 3 foods-13-03722-t003:** Thermal stability test results of the blended vegetable oils.

Oil Blend	Thermal Stability Tests	
	Rancimat Test, IP (Hours)	RapidOxy Test, IP (Minutes)
100S/0F	9.48 ± 0.14 ^k^	34.37 ± 0.46 ^k^
90S/10F	9.05 ± 0.15 ^j^	32.56 ± 0.52 ^j^
80S/20F	8.46 ± 0.12 ^i^	31.27 ± 0.34 ^i^
70S/30F	7.92 ± 0.08 ^h^	29.31 ± 0.46 ^h^
60S/40F	7.41 ± 0.07 ^g^	27.88 ± 0.27 ^g^
50S/50F	6.93 ± 0.06 ^f^	26.35 ± 0.32 ^f^
40S/60F	6.38 ± 0.08 ^e^	24.77 ± 0.22 ^e^
30S/70F	5.83 ± 0.05 ^d^	23.19 ± 0.36 ^d^
20S/80F	5.25 ± 0.11 ^c^	21.69 ± 0.27 ^c^
10S/90F	4.77 ± 0.09 ^b^	20.00 ± 0.33 ^b^
0S/100F	4.28 ± 0.08 ^a^	18.39 ± 0.36 ^a^
Control	16.47 ± 0.23 ^l^	70.50 ± 0.84 ^l^

Values are means ± standard deviation (n = 3). Different lower-case letters in the same column indicate significantly different values (*p* < 0.05), according to post hoc Tukey’s HSD test.

**Table 4 foods-13-03722-t004:** Color parameters of the blended vegetable oils (lightness L*, redness a*, yellowness b*, and total color differences ΔE).

Oil Blend	L*	a*	b*	ΔE
100S/0F	26.50 ± 0.01 ^j^	−1.00 ± 0.05 ^d^	3.50 ± 0.03 ^c^	1.28 ± 0.02 ^d^
90S/10F	25.42 ± 0.01 ^h^	−2.13 ± 0.10 ^b^	10.26 ± 0.03 ^b^	6.08 ± 0.02 ^h^
80S/20F	25.30 ± 0.02 ^g^	−1.70 ± 0.01 ^c^	11.25 ± 0.01 ^j^	7.02 ± 0.01 ^j^
70S/30F	24.97 ± 0.01 ^b^	−1.19 ± 0.02 ^a^	10.83 ± 0.02 ^i^	6.60 ± 0.00 ^i^
60S/40F	24.69 ± 0.00 ^a^	−0.77 ± 0.04 ^e^	10.58 ± 0.06 ^h^	6.40 ± 0.06 ^c^
50S/50F	24.85 ± 0.02 ^d^	−0.40 ± 0.06 ^f^	10.36 ± 0.06 ^ab^	6.19 ± 0.05 ^a^
40S/60F	24.66 ± 0.01 ^a^	−0.13 ± 0.07 ^g^	10.45 ± 0.04 ^a^	6.34 ± 0.02 ^bc^
30S/70F	25.00 ± 0.01 ^b^	0.07 ± 0.05 ^h^	10.39 ± 0.04 ^a^	6.28 ± 0.04 ^ab^
20S/80F	24.29 ± 0.01 ^c^	0.29 ± 0.02 ^i^	9.89 ± 0.04 ^g^	5.95 ± 0.04 ^g^
10S/90F	24.91 ± 0.01 ^e^	0.47 ± 0.03 ^j^	9.02 ± 0.04 ^f^	5.08 ± 0.03 ^f^
0S/100F	25.14 ± 0.01 ^f^	0.76 ± 0.03 ^k^	8.36 ± 0.04 ^e^	4.56 ± 0.03 ^e^
Control	25.48 ± 0.02 ^i^	−1.18 ± 0.06 ^a^	4.25 ± 0.02 ^d^	/

Values are means ± standard deviation (n = 3). Different lower-case letters in the same column indicate significantly different values (*p* < 0.05), according to post hoc Tukey’s HSD test.

**Table 5 foods-13-03722-t005:** Sensory attributes of the blended vegetable oils.

Oil Blend	Color	Odor	Taste	Average Rating	Total Acceptability
100S/0F	3.3 ± 0.9 ^b^	3.9 ± 1.2 ^a^	4.1 ± 1.4 ^a^	3.8 ± 0.9 ^a^	3.8 ± 1.0 ^a^
90S/10F	4.4 ± 0.7 ^a^	3.6 ± 1.0 ^a^	4.1 ± 1.0 ^a^	4.0 ± 0.6 ^a^	4.0 ± 0.7 ^a^
80S/20F	4.4 ± 0.7 ^a^	3.6 ± 0.8 ^a^	3.9 ± 1.1 ^ab^	4.0 ± 0.7 ^a^	3.9 ± 0.7 ^a^
70S/30F	4.1 ± 0.7 ^ab^	3.7 ± 0.8 ^a^	3.5 ± 1.0 ^ab^	3.8 ± 0.7 ^a^	3.7 ± 0.7 ^a^
60S/40F	4.1 ± 0.6 ^ab^	3.7 ± 0.7 ^a^	3.0 ± 1.1 ^ab^	3.6 ± 0.6 ^a^	3.5 ± 0.6 ^a^
50S/50F	3.8 ± 0.8 ^ab^	3.7 ± 0.9 ^a^	2.7 ± 1.3 ^ab^	3.4 ± 0.7 ^a^	3.3 ± 0.8 ^a^
40S/60F	3.9 ± 0.7 ^ab^	4.1 ± 1.0 ^a^	2.4 ± 1.5 ^ab^	3.5 ± 0.9 ^a^	3.3 ± 1.0 ^a^
30S/70F	3.7 ± 1.1 ^ab^	3.9 ± 0.7 ^a^	2.5 ± 1.7 ^ab^	3.4 ± 0.9 ^a^	3.3 ± 1.0 ^a^
20S/80F	3.9 ± 1.1 ^ab^	4.1 ± 0.7 ^a^	2.3 ± 1.9 ^b^	3.4 ± 0.8 ^a^	3.3 ± 0.9 ^a^
10S/90F	3.8 ± 0.9 ^ab^	4.2 ± 0.5 ^a^	2.3 ± 1.9 ^ab^	3.4 ± 0.7 ^a^	3.3 ± 0.9 ^a^
0S/100F	3.9 ± 1.0 ^ab^	3.9 ± 0.7 ^a^	2.6 ± 2.0 ^ab^	3.5 ± 0.9 ^a^	3.4 ± 1.0 ^a^
Control	3.6 ± 0.9 ^ab^	4.1 ± 1.2 ^a^	3.9 ± 1.1 ^ab^	3.9 ± 0.9 ^a^	3.9 ± 0.9 ^a^

Values are means ± standard deviation (n = 215); Different lower-case letters in the same column indicate significantly different values (*p* < 0.05), according to post hoc Tukey’s HSD test.

**Table 6 foods-13-03722-t006:** Acid value (AV), peroxide value (PV), and anisidine value.

Network Name	Performance			Error			Train. Algorith.	Error Func.	Hidden Active.	Output Active.
	Train.	Test.	Valid.	Train.	Test.	Valid.				
MLP 3-7-1	0.959	0.750	0.964	6.361	9.975	6.961	BFGS 67	SOS	Exponential	Identity

## Data Availability

The original contributions presented in the study are included in the article/Appendix A; further inquiries can be directed to the corresponding author.

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
