# Peer review of "Omega 3 Blends of Sunflower and Flaxseed Oil—Modeling Chemical Quality and Sensory Acceptability"

_foods, 2024, doi:10.3390/foods13233722_

Round 1

Reviewer 1 Report

Comments and Suggestions for Authors

This manuscript describes the blending of sunflower oil and falxseed oil as essential fatty acid sources to obtain a stable and fatty acid balnced formulations. It is an interesting subject as there is high content of omega 6 fatty acids in the daily diet and should be omega 3 fatty acids also involved in our diet as well to prevent many diseases.

There is much focus on the control sample obtained from the market, which should be avoided and the focus should be on the oil samples prepared in this research so revise the abstract and present advantage of your own blended samples.

As the aim of this study was to reach an oil with balanced essential fatty acids, please make a conclusion and introduce the best sample for readers at the end of abstract.

Please compare the PV and AV of the oil samples with the international standards such as CODEX as the obtained samples are consumable or not.

I would suggest to explain why there is no storage experiments?

In tables, first column, oil blend ratios was stated without defining that numbers in the ratio are related to which oils!

There is not deep discussion with already published scientific papers, for example: Chemical, Rheological and Nutritional Characteristics of Sesame and Olive Oils Blended with Linseed Oil

Adv Pharm Bull 8 (1), 107-113

I would suggest to add a column in the Table showing the omega 6/omega 3 ratio for all the blends.

Please delete data for C14:0 from the table as it is not present in much amount and is ND for many of the samples.

Author Response

This manuscript describes the blending of sunflower oil and falxseed oil as essential fatty acid sources to obtain a stable and fatty acid balnced formulations. It is an interesting subject as there is high content of omega 6 fatty acids in the daily diet and should be omega 3 fatty acids also involved in our diet as well to prevent many diseases.

AUTHORS: The authors are grateful to the reviewer for the comments and observations.

There is much focus on the control sample obtained from the market, which should be avoided and the focus should be on the oil samples prepared in this research so revise the abstract and present advantage of your own blended samples.

AUTHORS: Thank you for bringing the control sample to our attention. The authors modified the abstract by emphasizing the tested oil blends.

Lines 15-26: Principal Component Analysis (PCA) showed that the optimal fatty acid composition was achieved in the sample with 20% sunflower oil and 80% flaxseed oil (20S/80F). However, developing a new product is highly complex due to the importance of oil quality and sensory charac-teristics. Therefore, an Artificial Neural Network (ANN) was applied to optimize the proportions of flaxseed and sunflower oil to create an oil blend with improved nutritional, oxidative, and sen-sory characteristics compared to the individual oils. The ANN analysis determined the optimal composition of the oil blend to be 51.5% refined sunflower oil and 48.5% cold-pressed flaxseed oil. Sensory characteristics pose a particular challenge in optimization, as flaxseed oil, which increases essential omega 3 fatty acids, has a specific taste that is not widely favored by consumers. None-theless, by blending with refined sunflower oil, the resulting optimal blend (51.5% refined sun-flower oil and 48.5% cold-pressed flaxseed oil) possesses pleasant sensory characteristics.

As the aim of this study was to reach an oil with balanced essential fatty acids, please make a conclusion and introduce the best sample for readers at the end of abstract.

AUTHORS: Thank you for the observation. If only the fatty acid composition is considered, the Principal Component Analysis showed that the sample with 80% refined sunflower oil and 20% cold-pressed flaxseed oil was the best. However, it is important to take into account the chemical quality and sensory characteristics of the oil, and for that reason optimization was done using Artificial Neural Network and an oil with an optimal composition of 51.5% refined sunflower seed oil and 48.5% of cold-pressed flaxseed oil was obtained.

The authors modified the abstract according to reviewer´s recommendations.

Please compare the PV and AV of the oil samples with the international standards such as CODEX as the obtained samples are consumable or not.

AUTHORS: PV and AV of all investigated samples are in line with CODEX. In the sections 3.2.1. AV and 3.2.2. PV, AnV and TOTOX following was added:

Lines 330-331: The acid values obtained in all investigated samples are in accordance with the Codex Alimentarius Standards, 1999 [51].

Lines 345-346: Obtained results were in line with previous studies [56,58] and Codex Alimentarius Standards, 1999 [51] for PV.

I would suggest to explain why there is no storage experiments?

AUTHORS: Thank you for this comment. The authors agree with the reviewer that the results of the storage test are missing from the manuscript. Since the storage test at room temperature takes a long time due to the long shelf life of the oil, the authors decided to perform accelerated stability tests (Rancimat and RapidOxy). The authors hope that the results of these tests will be satisfactory to the reviewer. Accelerated stability tests have been added to the 2. Materials and Methods, as well as to the 3. Results and Discussion section.

2.4. Accelerated stability tests

            In order to investigate oxidative stability of samples, two accelerated stability tests were performed: Rancimat and RapidOxy test.

2.4.1. Rancimat test

Rancimat accelerated stability test was made on Rancimat apparatus, model 743 (Metrohm, Herisau, Switzerland) to measure the induction period of oil samples. Measurements were performed according to ISO 6886, 2016 [38]. The procedure and measurement conditions were previously described by Lužaić et al., 2022 [39]. Namely, 2.50 ± 0.01 g of oil sample was oxidized at a temperature of 100°C and the airflow of 18-20 L/h. Volatile products formed during the oxidation were diluted in 0.05 L of distilled water. The device measures the conductivity recorded by apparatus software and is used to obtain the induction period (IP) expressed in hours with an accuracy of 0.1.

2.4.2. RapidOxy test

     The oxidative stability of the oil was also tested using RapidOxy 100 (Anton Paar, Germany). Namely, 3.00 ± 0.01 g of the sample was weighed into a glass container and placed in the device chamber. Oxygen was introduced into the chamber until the pressure rose to 700 kPa. Subsequently, the chamber with the sample was heated to 140°C. As the temperature increased, the pressure in the chamber also increased (up to about 1000 kPa) until oxidation began. As a result of the reaction of oxygen with the tested oil sample, oxygen was consumed, and the pressure inside the chamber decreased. When the pressure in the chamber dropped by 10%, the oxidation reaction was considered complete, and the time was recorded as an induction period (IP) in minutes [40].

3.3. Accelerated stability tests

The oxidative stability of the oils was investigated using accelerated thermal stability tests. Similar results were obtained using the Rancimat and RapidOxy tests, as shown in Table 3. Refined sunflower oil displayed good oxidative stability characteristics with an induction period of 9.48 ± 0.14 hours obtained by the Rancimat test, or 34.37 ± 0.46 minutes by the RapidOxy test. These results are consistent with previous research reported by Velasco et al., 2004 [62] and Lužaić et al., 2022 [39]. With addition of cold-pressed flaxseed oil, the induction period was significantly reduced to 4.28 ± 0.08 h (Rancimat test) or 18.39 ± 0.36 minutes (RapidOxy test), as determined in the pure cold-pressed flaxseed oil. Similar values were previously reported by TaÅ„ska et al. 2016 [63] and MikoÅ‚ajczak and TaÅ„ska, 2022 [64]. Significantly higher induction period values were observed in the control sample (16.47 ± 0.23 hours and 70.50 ± 0.84 minutes), which is a consequence of the different composition of fatty acids (essential and non-essential), the content and composition of minor components with pro-oxidative and antioxidant effects, the production process, raw materials, etc. [65,66]. Oxidative changes occur at the unsaturated bonds of fatty acids, so the influence of the polyunsaturated fatty acids (PUFA) content and the induction period obtained by accelerated stability tests was examined, revealing an extremely strong negative correlation between the PUFA content (based on the results, Table 1) and the induction period as determined by the Rancimat test (R=-0.984, p=0.000) and the RapidOxy test (R=-0.998, p=0.000) in the investigated samples of blended vegetable oils.

Table 3. Thermal stability test results of the blended vegetable oils

Oil                          blend

Thermal stability tests

Rancimat test, IP (hours)

RapidOxy test, IP (minutes)

100S/0F

9.48 ± 0.14k

34.37 ± 0.46k

90S/10F

9.05 ± 0.15j

32.56 ± 0.52j

80S/20F

8.46 ± 0.12i

31.27 ± 0.34i

70S/30F

7.92 ± 0.08h

29.31 ± 0.46h

60S/40F

7.41 ± 0.07g

27.88 ± 0.27g

50S/50F

6.93 ± 0.06f

26.35 ± 0.32f

40S/60F

6.38 ± 0.08e

24.77 ± 0.22e

30S/70F

5.83 ± 0.05d

23.19 ± 0.36d

20S/80F

5.25 ± 0.11c

21.69 ± 0.27c

10S/90F

4.77 ± 0.09b

20.00 ± 0.33b

0S/100F

4.28 ± 0.08a

18.39 ± 0.36a

Control sample

16.47 ± 0.23l

70.50 ± 0.84l

Values are means ± standard deviation (n=3).

Different lower-case letters in the same column indicate significantly different values (p<0.05), according to post hoc Tukey's HSD test.

  1. ISO 6886:2016. Animal and vegetable fats and oils - Determination of oxidative stability (accelerated oxidation test). International Organization for Standardization, Geneva, Switzerland.
  2. Lužaić, T. Z., Grahovac, N. L., Hladni, N. T., Romanić, R. S. Evaluation of oxidative stability of new cold-pressed sunflower oils during accelerate thermal stabil-ity tests. Food Sci. Technol. (Campinas) 2022, 42,
  3. Anton Paar. Determination of Oxidation Stability – a User Guideline. Relevant for: Food Industry; Anton Paar: Graz, Austria, 2022.
  4. Velasco, J., Andersen, M., Skibsted, L. Evaluation of oxidative stability of vegetable oils by monitoring the tendency to radical formation. A comparison of electron spin resonance spectroscopy with the Rancimat method and differential scanning calorimetry. Food Chem. 2004, 85, 623-632.
  5. TaÅ„ska, M., Roszkowska, B., Skrajda, M., DÄ…browski, G. Commercial cold pressed flaxseed oils quality and oxidative stability at the beginning and the end of their shelf life. Oleo Sci. 2016, 65, 111–121.
  6. MikoÅ‚ajczak, N., TaÅ„ska, M. Effect of initial quality and bioactive compounds content in cold-pressed flaxseed oils on oxidative stability and oxidation products formation during one-month storage with light exposure. NFS Journal 2022, 26, 10–21.
  7. Choe, E., Min, D. B. Mechanisms and factors for edible oil oxidation. CRFSFS 2006, 5, 169-186.
  8. Sabolová, M., Zeman, V., Lebedová, G., Doležal, M., Soukup, J., Réblová, Z. Relationship between the fat and oil composition and their initial oxidation rate during storage. Czech J. Food Sci. 2020, 38, 404–409.

In tables, first column, oil blend ratios was stated without defining that numbers in the ratio are related to which oils!

AUTHORS: The authors agree that it is not clear in the table itself which oil the ratio refers to, so the authors have corrected this in all tables.

There is not deep discussion with already published scientific papers, for example: Chemical, Rheological and Nutritional Characteristics of Sesame and Olive Oils Blended with Linseed Oil

Adv Pharm Bull 8 (1), 107-113

AUTHORS: The authors agree with the reviewer's comments. The manuscript is supplemented by an in-depth discussion and comparison of the obtained results with already published scientific papers.

Lines 265-271: Similar conclusions were reached by Hashempour-Baltork et al., 2018 [50], by blending olive, flaxseed, and sesame oils in different proportions. Flaxseed oil was used as a source of essential omega 3 fatty acids, while sesame oil was a source of essential omega 6 fatty acids. With the increase in the proportion of flaxseed oil, the content of omega 3 fatty acids increased, while with the increase in the proportion of sesame oil, the content of omega 6 fatty acids increased, and vice versa.

Lines 328-331: Grover et al., 2021 [59], reported similar values blending flaxseed and sunflower oil in various proportions (80:20, 70:30, 60:40, 50:50, 40:60, 70:30, 80:20), determining AVs ranging from 1.07 ± 0.10 to 1.12 ± 0.10 mgKOH/g. The acid values obtained in all investigated samples are in accordance with the Codex Alimentarius Standards, 1999 [51].

  1. Hashempour-Baltork, F., Torbati, M., Azadmard-Damirchi, S., Savage, G. P. Chemical, rheological and nutritional characteristics of sesame and olive oils blended with linseed oil. Pharm. Bull. 2018, 8, 107–113.
  2. Codex Alimentarius, 1999. Standard for named vegetable oils Codex Stan 210-1999. Codex Alimentarius.
  3. Grover, S., Kumari, P., Kumar, A., Soni, A., Sehgal, S., Sharma, V. Preparation and Quality Evaluation of Different Oil Blends. Letters in Applied NanoBioScience, 2021, 10, 2126–2137.

I would suggest to add a column in the Table showing the omega 6/omega 3 ratio for all the blends.

AUTHORS: The results of the omega 6/omega 3 ratio, the indices of atherogenicity (IA) and thrombogenicity (IT) as well as hypocholesterolaemic/hypercholesterolaemic ratio have been added to Table 1. Also, subtitle 3.1. Fatty acid composition of the 3. Results and Discussion section was supplemented with a discussion of the results:

Lines 282-315: Investigating the ω6/ω3 fatty acid ratio is significant from a health perspective, and studying the ideal ratio is the subject of many scientific papers. Retrospectively, this ratio has increased multiple times from 1 to over 20 due to a diet rich in red meat, dairy products, and salt, while being low in fruits, vegetables, legumes, and fish [22]. Different ratios are important in the prevention of certain diseases. For example, a ratio of 4 is associated with reduced mortality from cardiovascular diseases, a ratio of 2-3 is significant for patients with colorectal tumors and rheumatoid arthritis, and a ratio of 5 has a beneficial effect for patients suffering from asthma [22,52]. To achieve a balanced ω6/ω3 fatty acid ratio, lower ratios should be aimed for [53,54], and it is considered that a value range of 1 to 5 is optimal for human health [52]. Among the samples analyzed, the extremely high value (892.91) is characteristic of pure refined sunflower oil, while even a minimal addition of flaxseed oil significantly reduces the ratio (to 13.19). Optimal values of this ratio (1-5) were achieved by adding 30 to 60% cold-pressed flaxseed oil. In the control sample, a ratio of 4.96 was determined, which is close to the maximum recommended ratio, while the addition of 20% flaxseed oil resulted in a ratio slightly higher than the recommended (5.57).

To assess the nutritional quality of mixed oils, nutritional indices including IA, IT, and HH were calculated. The index of atherogenicity (IA) is based on the ratio of saturated to unsaturated fatty acids describes the atherogenic potential of fatty acids. Consuming foods with lower IA can contribute to reducing total and LDL cholesterol levels [55]. The index of thrombogenicity (IT) characterizes the thrombogenic potential of fatty acids and represents the relationship between pro-thrombogenic (C14:0, C16:0, and C18:0) and anti-thrombogenic fatty acids (monounsaturated, ω3, and ω6 fatty acids) [29]. Essentially, both indices are associated with cardiovascular disease risk and must be as low as possible.

The obtained IA and IT values in all oil samples are far below 1. It should be noted that with the change in fatty acid composition resulting from the addition of flaxseed oil, IA values vary slightly (0.06-0.07), while IT values decrease (0.19-0.05). The lowest IA value was obtained for control sample at 0.04. Considering the hypocholesterolemic/hypercholesterolemic fatty acid ratio [30], the HH index considers the effect of fatty acids on cholesterol metabolism, and from a nutritional standpoint, high values of this index are desirable. Enriching sunflower oil with flaxseed oil increases the HH index in the analyzed samples from 14.96 to 18.03. The highest value of this index (23.26) was determined in control sample.

  1. Simopoulos, A. P. An increase in the omega-6/omega-3 fatty acid ratio increases the risk for obesity. Nutrients 2016, 8, 128-145.
  2. Ubricht, T.L.V.; Southgate, D.A.T. Coronary heart disease: Seven dietary factors. Lancet 1991, 338, 985–992.
  3. Santos-Silva, J., Bessa, R. J. B., Santos-Silva, F. Effect of genotype, feeding system and slaughter weight on the quality of light lambs: II. Fatty acid composition of meat. Prod. Sci. 2002, 77, 187-194.
  4. Lupette, J., Benning, C. Human health benefits of very-long-chain polyun-saturated fatty acids from microalgae. Biochimie 2020, 178, 15-25.
  5. Opinion of the French Food Safety Agency(AFSSA)on the up-date of French population reference intakes(ANCs)for fatty acids. 2010. Available online: https://www.anses.fr/fr/system/files/NUT2006sa0359.pdf (accessed on 10.10.2022).
  6. Simopoulos, A. P. The omega-6/omega-3 fatty acid ratio: health implications. OCL 2010, 17, 267-275.
  7. Yurchenko, S., Sats, A., Tatar, V., Kaart, T., Mootse, H., Jõudu, I. Fatty acid profile of milk from Saanen and Swedish Landrace goats. Food Chem. 2018, 254, 326-332.

Please delete data for C14:0 from the table as it is not present in much amount and is ND for many of the samples.

AUTHORS: Thank you for this comment. Authors deleted C14:0 colomn in the Table 1.

Reviewer 2 Report

Comments and Suggestions for Authors

 A thorough revision and improvement of the manuscript would be needed to meet the quality standards of a first quartile Q1 journal, because, in addition to the below-mentioned English language improvement:

- The introduction contains sentences copied almost literally from the review of Hashempour-Baltork et al., (2016) (https://doi.org/10.1016/j.tifs.2016.09.007), which is unacceptable. Besides, it is poor and with unconnected ideas.  

-Important conceptual errors are observed, as for example:

L28: "Oil quality" encompasses all the properties indicated behind (stability...).

The term fortified is used erroneously, since making an oil blend is not fortifying the oil. It is simply a blend. Fortification implies the addition of micronutritienst to increase food nutritional value, such as universal salt iodization or the fortification of maize flour, corn meal, wheat flour and rice with vitamins and minerals.

-The results are very reminiscent of a previous article by the same authors (Romanic et al, 2021, https://doi.org/10.1016/j.lwt.2021.1120), but are less well discussed.

-Supplementary data is missing

-Units are also missing in Table 1 and also in different parts of the discussion, such as L394-404, L424-427...

Comments on the Quality of English Language

 Extensive editing of English language is required, because there are several mistakes along the text. For example:

L13 "reach" instead of "rich"

L17, L21: most like?

L209: "Of the polyunsaturated..."

L432: "can enable"?

Author Response

A thorough revision and improvement of the manuscript would be needed to meet the quality standards of a first quartile Q1 journal, because, in addition to the below-mentioned English language improvement:

AUTHORS: Authors would like to thank the reviewer for the comments and the opportunity to try to address all the mentioned shortcomings. Also, the paper was thoroughly reviewed by a native English speaker and changes were marked in the manuscript.

- The introduction contains sentences copied almost literally from the review of Hashempour-Baltork et al., (2016) (https://doi.org/10.1016/j.tifs.2016.09.007), which is unacceptable. Besides, it is poor and with unconnected ideas.  

AUTHORS: Thak you for this observation. The authors removed the disputed sentence and modified the 1. Introduction section according to reviewers comment.

Lines 34-51: Vegetable oil is an essential component of the diet and represents the primary source of lipids, substances that not only provide energy but also contribute to the construction of lipid membranes in the body [1,2]. It is a significant source of fatty acids, especially essential ones, and also facilitates the absorption of liposoluble vitamins. In human nutrition, vegetable oils are most commonly used for cooking and are extensively utilized in the food industry, where their composition and quality in terms of nutritional and sensory characteristics define the quality of the final food product [3]. Currently, no single vegetable oil can be characterized by ideal functional, nutritional, and sensory characteristics along with adequate oxidative stability [4]. The nutritional quality and stability of vegetable oils can be improved through hydrogenation, interesterification, fractionation, and blending. Most of these processes come with certain limitations, including the need for expensive equipment and substantial financial investments, while hydrogenation has the drawback of forming harmful trans isomers. Conversely, blending is an efficient method for producing a functional product – a blended oil with a balanced fatty acid content and positive health effects [4,5]. Blending vegetable oils with different properties is one of the very simple ways to create new specific products with desired sensory, physicochemical, and oxidative properties [6]. Recent literature on blended oils with balanced fatty acid composition is predominantly limited to Asian countries, where blended oils based on rapeseed oil, palm oil, flaxseed oil, as well as sunflower and rice bran oils dominate [7-12], i.e., oils from oilseeds characteristic for specific geographic areas.

Lines 73-88: Determining the ω6/ω3 fatty acid ratio is significant from a health perspective, and studying the ideal ratio is the subject of many scientific papers. Retrospectively, this ratio has increased multiple times from 1 to over 20 due to a diet rich in red meat, dairy products, and salt, while being low in fruits, vegetables, legumes, and fish [22]. Depending on dietary habits, traditions, and food availability, the intake of essential fatty acids varies worldwide [23]. Stark et al., 2016 [23] and Schuchardt et al., 2024 [24] created a map of essential omega-3 fatty acid intake globally, finding that countries like Japan, South Korea, some Scandinavian countries (Norway and Finland), and territories such as Greenland (Denmark) and Alaska (USA) have exceptionally high intakes of these fatty acids. In contrast, countries like Brazil, Egypt, Iran, and India have very low intakes. European countries, the USA, Canada, China, and Australia have moderate intakes. Given the diversity in the intake of essential fatty acids, recommendations vary by geographic area. However, since most populations face an omega-3 fatty acid deficiency, the World Health Organization (WHO) has recommended a balanced intake of essential omega-6 (linoleic acid) and omega-3 (alpha-linolenic acid) in a ratio between 5:1 and 10:1 [25].

  1. Xu, H.; Zhu, L.; Dong, J.; Wei, Q.; Lei, M. Composition of Catalpa ovata Seed Oil and Flavonoids in Seed Meal as Well as Their Antioxidant Activities. Am. Oil Chem. Soc. 2015, 92, 361–369.
  2. Tian, M., Bai, Y., Tian, H., Zhao, X. The Chemical Composition and Health-Promoting Benefits of Vegetable Oils—A Review. Molecules, 2023, 28, 6393.
  3. Gunstone, F. D. Vegetable Oils in Food Technology: Composition, Properties and Uses, 2nd ed.; John Wiley and Sons, 2011.
  4. Hashempour-Baltork, F., Torbati, M., Azadmard-Damirchi, S., Savage, G.P. Vegetable oil blending: A review of physicochemical, nutritional and health effects. Trends Food Sci. Tech. 2016, 57, 52-58.
  5. O’Brien, R.D. Fats and oils: formulating and processing for applications, 3rd ed.; CRC Press, New York, USA, 2009; pp. 1-574.
  6. Hamed, S. F., Abo-Elwafa, G. A. Enhancement of oxidation stability of flax seed oil by blending with stable vegetable oils. J. Appl. Sci. 2012, 8, 5039–5048.
  7. Umesha, S. S., Naidu, K. A. Vegetable oil blends with α-linolenic acid rich Garden cress oil modulate lipid metabolism in experimental rats. Food Chem. 2012, 135, 2845–2851.
  8. Choudhary, M., Grover, K., Kaur, G. Fatty acid composition, oxidative stability, and radical scavenging activity of rice bran oil blends. J. Food Sci. Nutr. 2013, 2, 33-43.
  9. Reddy, K. J., Jayathilakan, K., Pandey, M. C., Radhakrishna, K. Evaluation of the physico-chemical stability of rice bran oil and its blends for the development of functional meat products. IJFANS 2013, 2, 46-53.
  10. Sharma, M., Lokesh, B. R. Modification of serum and tissue lipids in rats fed with blended and interesterified oils containing groundnut oil with linseed oil. Food Biochem. 2013, 37, 220-230.
  11. Adeyemi, K. D., Sazili, A. Q., Ebrahimi, M., Samsudin, A. A., Alimon, A. R., Karim, R., Karsani, S. A. Sabow, A. B. Effects of blend of canola oil and palm oil on nutrient intake and digestibility, growth performance, rumen fermentation and fatty acids in goats. Sci. J. 2016, 87, 1137–1147.
  12. Nehdi, I. A., Hadj-Kali, M. K., Sbihi, H. M., Tan, C. P., Al-Resayes, S. I. Characterization of ternary blends of vegetable oils with optimal ω-6/ω-3 fatty acid ratios. J. Oleo Sci. 2019, 68, 1041-1049.
  13. Simopoulos, A. P. An increase in the omega-6/omega-3 fatty acid ratio increases the risk for obesity. Nutrients 2016, 8, 128-145.
  14. Stark, K. D., Van Elswyk, M. E., Higgins, M. R., Weatherford, C. A., Salem, N. Global survey of the omega-3 fatty acids, docosahexaenoic acid and eicosapen-taenoic acid in the blood stream of healthy adults. Lipid Res. 2016, 63, 132-152.
  15. Schuchardt, J.P., Beinhorn, P., Hu, X.F., Chan, H.M., Roke, K., Bernasconi, A., Hahn, A., Sala-Vila, A., Stark, K.D., Harris, W.S. Omega-3 world map: 2024 update. Lipid Res. 2024, 95, 101286.

- Important conceptual errors are observed, as for example:

L28: "Oil quality" encompasses all the properties indicated behind (stability...).

AUTHORS: Thank you for pointing out this omission. The sentence has been corrected by removing the „stability” word.

The term fortified is used erroneously, since making an oil blend is not fortifying the oil. It is simply a blend. Fortification implies the addition of micronutritienst to increase food nutritional value, such as universal salt iodization or the fortification of maize flour, corn meal, wheat flour and rice with vitamins and minerals.

AUTHORS: The authors agree with the criticism of the reviewer. The word „fortified” in the manuscript was replaced by the word „enriched“.

- The results are very reminiscent of a previous article by the same authors (Romanic et al, 2021, https://doi.org/10.1016/j.lwt.2021.1120), but are less well discussed.

AUTHORS: The results of this paper are part of the research presented in the article Romanić et al. (2021). The mentioned article emphasized the nutritional characteristics of blended vegetable oils. The other results, not included in the article, are highly important. So authors decided to present these results in this manuscript (fatty acids composition, oil quality parameters, instrumental color measurement, sensory acceptability, and based on the reviewers' comments authors added nutritient indices calculated based on the fatty acid composition and stability tests - Rancimat and RapidOxy in the R1 version). In the R1 version of the manuscript sections 2. Material and Methods (2.2. Fatty acid composition and 2.4. Accelerated stability tests), as well as 3. Results and Discussion section (3.1. Fatty acid composition and 3.3. Accelerated stability tests) are supplemented with mentioned results. The discussion of the results is also deepened in the R1 version of the manuscript (see 3. Results and Discussion section).

- Supplementary data is missing

AUTHORS: Thank you for pointing out this shortcoming. Supplementary material will be added in R1 version.

- Units are also missing in Table 1 and also in different parts of the discussion, such as L394-404, L424-427...

AUTHORS: Thank you for this comment. Autors added units in Table 1, also unites were checked and  added in the whole manuscript.

Comments on the Quality of English Language

Extensive editing of English language is required, because there are several mistakes along the text. For example:

L13 "reach" instead of "rich"

L17, L21: most like?

L209: "Of the polyunsaturated..."

L432: "can enable"?

AUTHORS: Thank you for this comment. All mentioned mistakes were corrected. Also, the manuscript was thoroughly reviewed by a native English speaker and changes were marked in the manuscript.

Reviewer 3 Report

Comments and Suggestions for Authors

Dear Authors,

your manuscript titled "Omega 3 Blends of Sunflower and Flaxseed Oil - Modeling Chemical Quality and Sensory Acceptability" has been submitted for my review. In this work, you analyze different blends of oils aiming to achieve both higher nutritional value and sensory appeal. Although you have conducted thorough research, the article contains certain shortcomings and ambiguities. I have a few comments for your consideration:

  1. In the introduction, the authors only briefly mention, in the final paragraph, the recommendations regarding n3 and n6 fatty acid intake. However, it is not clarified that these recommendations vary significantly across countries and are influenced by the consumption of traditional dishes in each. Moreover, there is no mention that the vast majority of foods exhibit a very unfavorable n6/n3 ratio, often exceeding 20:1.
  2. There is no reference to many essential publications on oil blends, which have already been widely discussed in the literature.
  3. While the article mentions the ratio of n3 to n6 fatty acids, there is a lack of nutritional context or discussion of their health impacts. Including studies on the health benefits of an appropriate n3/n6 ratio could significantly enhance the discussion section. I also suggest calculating and adding nutritional value indicators for the oils, with methodologies that are readily available in the literature, such as in: 10.3390/foods12142660 or 10.3390/nu14183795.
  4. The article applies ISO standards to assess the quality of oils, such as peroxide and acid values, but lacks complete explanations as to why these indicators are critical in the context of the analyzed blends.
  5. Although oxidative stability is analyzed through parameters such as TOTOX, the article does not fully present results related to long-term storage and quality changes. Studies simulating storage conditions or at least conducting thermal stability tests would be recommended.
  6. Sensory results indicate the impact of flaxseed oil on the taste of the mixtures; however, there is a lack of deeper analysis of this phenomenon. It would be beneficial to discuss possible chemical reactions responsible for the negative taste perception at higher flaxseed oil contents.
  7. The use of statistical methods such as PCA and ANN requires additional clarification, especially regarding the influence of individual components on blend optimization. The ANN analysis is not fully transparent, particularly in terms of the selection of weight parameters and their justification. It would be useful to explain how ANN contributed to determining the optimal ratio of sunflower and flaxseed oils.

The article provides valuable data; however, it requires additions and clarifications before proceeding to further stages.

Comments on the Quality of English Language

Minor editing of English language required.

Author Response

Dear Authors,

your manuscript titled "Omega 3 Blends of Sunflower and Flaxseed Oil - Modeling Chemical Quality and Sensory Acceptability" has been submitted for my review. In this work, you analyze different blends of oils aiming to achieve both higher nutritional value and sensory appeal. Although you have conducted thorough research, the article contains certain shortcomings and ambiguities. I have a few comments for your consideration:

AUTHORS: Thank you for valuble comments. Authors will try to improve the manuscript acoording to reviewer´s observations.

  1. In the introduction, the authors only briefly mention, in the final paragraph, the recommendations regarding n3 and n6 fatty acid intake. However, it is not clarified that these recommendations vary significantly across countries and are influenced by the consumption of traditional dishes in each. Moreover, there is no mention that the vast majority of foods exhibit a very unfavorable n6/n3 ratio, often exceeding 20:1.

AUTHORS: Authors previosly published the paper regarding the nutritional characteristics of blended vegetable oils (Romanić et al, 2021, https://doi.org/10.1016/j.lwt.2021.1120). The mentioned article discussed in detail the recommendation for the intake of omega 3 and omega 6 fatty acids.  This manuscript emphasizes chemical quality and sensory acceptability of the investigated oil blends. Therefore the authors in this manuscript did not deal in detail with the nutritional characteristics. However, as the reviewer considers it important, the authors have added an explanation of omega 3 and omega 6 fatty acid intake recommendations to the 1. Introduction section.

Lines 73-88: Determining the ω6/ω3 fatty acid ratio is significant from a health perspective, and studying the ideal ratio is the subject of many scientific papers. Retrospectively, this ratio has increased multiple times from 1 to over 20 due to a diet rich in red meat, dairy products, and salt, while being low in fruits, vegetables, legumes, and fish [22]. Depending on dietary habits, traditions, and food availability, the intake of essential fatty acids varies worldwide [23]. Stark et al., 2016 [23] and Schuchardt et al., 2024 [24] created a map of essential omega-3 fatty acid intake globally, finding that countries like Japan, South Korea, some Scandinavian countries (Norway and Finland), and territories such as Greenland (Denmark) and Alaska (USA) have exceptionally high intakes of these fatty acids. In contrast, countries like Brazil, Egypt, Iran, and India have very low intakes. European countries, the USA, Canada, China, and Australia have moderate intakes. Given the diversity in the intake of essential fatty acids, recommendations vary by geographic area. However, since most populations face an omega-3 fatty acid deficiency, the World Health Organization (WHO) has recommended a balanced intake of essential omega-6 (linoleic acid) and omega-3 (alpha-linolenic acid) in a ratio between 5:1 and 10:1 [25].

  1. Simopoulos, A. P. An increase in the omega-6/omega-3 fatty acid ratio increases the risk for obesity. Nutrients 2016, 8, 128-145.
  2. Stark, K. D., Van Elswyk, M. E., Higgins, M. R., Weatherford, C. A., Salem, N. Global survey of the omega-3 fatty acids, docosahexaenoic acid and eicosapen-taenoic acid in the blood stream of healthy adults. Lipid Res. 2016, 63, 132-152.
  3. Schuchardt, J.P., Beinhorn, P., Hu, X.F., Chan, H.M., Roke, K., Bernasconi, A., Hahn, A., Sala-Vila, A., Stark, K.D., Harris, W.S. Omega-3 world map: 2024 update. Lipid Res. 2024, 95, 101286.
  4. There is no reference to many essential publications on oil blends, which have already been widely discussed in the literature.

AUTHORS: The authors agree with the reviewer's comments. The manuscript is supplemented by an in-depth discussion and comparison of the obtained results with already published scientific papers.

Lines 265-271: Similar conclusions were reached by Hashempour-Baltork et al., 2018 [50], by blending olive, flaxseed, and sesame oils in different proportions. Flaxseed oil was used as a source of essential omega-3 fatty acids, while sesame oil was a source of essential omega-6 fatty acids. With the increase in the proportion of flaxseed oil, the content of omega-3 fatty acids increased, while with the increase in the proportion of sesame oil, the content of omega-6 fatty acids increased, and vice versa.

Lines 328-331: Grover et al., 2021 [59], reported similar values blending flaxseed and sunflower oil in various proportions (80:20, 70:30, 60:40, 50:50, 40:60, 70:30, 80:20), determining AVs ranging from 1.07 ± 0.10 to 1.12 ± 0.10 mgKOH/g.

  1. Hashempour-Baltork, F., Torbati, M., Azadmard-Damirchi, S., Savage, G. P. Chemical, rheological and nutritional characteristics of sesame and olive oils blended with linseed oil. Pharm. Bull. 2018, 8, 107–113.
  2. Grover, S., Kumari, P., Kumar, A., Soni, A., Sehgal, S., Sharma, V. Preparation and Quality Evaluation of Different Oil Blends. Letters in Applied NanoBioScience, 2021, 10, 2126–2137.

  1. While the article mentions the ratio of n3 to n6 fatty acids, there is a lack of nutritional context or discussion of their health impacts. Including studies on the health benefits of an appropriate n3/n6 ratio could significantly enhance the discussion section. I also suggest calculating and adding nutritional value indicators for the oils, with methodologies that are readily available in the literature, such as in: 10.3390/foods12142660 or 10.3390/nu14183795.

AUTHORS: The authors agree with the reviewer's comment. The omega 6/omega 3 fatty acid ratio, the indices of atherogenicity (IA) and thrombogenicity (IT) as well as hypocholesterolaemic/hypercholesterolaemic ratio and discussion of the corresponding results have been added to the manuscript. Subtitle 2.2. Fatty acid composition of the 2. Materials and Methods section was supplemented:

Lines 122-130: The omega 6 and omega 3 (ω6/ω3) fatty acids ratio and nutritional indices, which assessed the nutritional quality of the analyzed oils, were determined based on the fatty acid composition, i.e., the content of individual fatty acids. The calculation for the indices of atherogenicity (IA) and thrombogenicity (IT) were developed by Ulbricht and Southgate, 1991 [29], while the hypocholesterolaemic/hypercholesterolaemic ratio was proposed by Santos-Silva et al., 2002 [30]. The indices were calculated based on the following Eq. 1-3:

                                                                                                                     (1)

                                                                                                          (2)

                   (3)

  1. Ubricht, T.L.V.; Southgate, D.A.T. Coronary heart disease: Seven dietary factors. Lancet 1991, 338, 985–992.
  2. Santos-Silva, J., Bessa, R. J. B., Santos-Silva, F. Effect of genotype, feeding system and slaughter weight on the quality of light lambs: II. Fatty acid composition of meat. Prod. Sci. 2002, 77, 187-194.

Nutrient indices results were added to Table 1, while subtitle 3.1. Fatty acid composition of section 3. Results and Discussion was supplemented with a discussion of presented results:

Lines 282-315: Investigating the ω6/ω3 fatty acid ratio is significant from a health perspective, and studying the ideal ratio is the subject of many scientific papers. Retrospectively, this ratio has increased multiple times from 1 to over 20 due to a diet rich in red meat, dairy products, and salt, while being low in fruits, vegetables, legumes, and fish [22]. Different ratios are important in the prevention of certain diseases. For example, a ratio of 4 is associated with reduced mortality from cardiovascular diseases, a ratio of 2-3 is significant for patients with colorectal tumors and rheumatoid arthritis, and a ratio of 5 has a beneficial effect for patients suffering from asthma [22,52]. To achieve a balanced ω6/ω3 fatty acid ratio, lower ratios should be aimed for [53,54], and it is considered that a value range of 1 to 5 is optimal for human health [52]. Among the samples analyzed, the extremely high value (892.91) is characteristic of pure refined sunflower oil, while even a minimal addition of flaxseed oil significantly reduces the ratio (to 13.19). Optimal values of this ratio (1-5) were achieved by adding 30 to 60% cold-pressed flaxseed oil. In the control sample, a ratio of 4.96 was determined, which is close to the maximum recommended ratio, while the addition of 20% flaxseed oil resulted in a ratio slightly higher than the recommended (5.57).

To assess the nutritional quality of mixed oils, nutritional indices including IA, IT, and HH were calculated. The index of atherogenicity (IA) is based on the ratio of saturated to unsaturated fatty acids describes the atherogenic potential of fatty acids. Consuming foods with lower IA can contribute to reducing total and LDL cholesterol levels [55]. The index of thrombogenicity (IT) characterizes the thrombogenic potential of fatty acids and represents the relationship between pro-thrombogenic (C14:0, C16:0, and C18:0) and anti-thrombogenic fatty acids (monounsaturated, ω3, and ω6 fatty acids) [29]. Essentially, both indices are associated with cardiovascular disease risk and must be as low as possible.

The obtained IA and IT values in all oil samples are far below 1. It should be noted that with the change in fatty acid composition resulting from the addition of flaxseed oil, IA values vary slightly (0.06-0.07), while IT values decrease (0.19-0.05). The lowest IA value was obtained for control sample at 0.04. Considering the hypocholesterolemic/hypercholesterolemic fatty acid ratio [30], the HH index considers the effect of fatty acids on cholesterol metabolism, and from a nutritional standpoint, high values of this index are desirable. Enriching sunflower oil with flaxseed oil increases the HH index in the analyzed samples from 14.96 to 18.03. The highest value of this index (23.26) was determined in control sample.

  1. Simopoulos, A. P. An increase in the omega-6/omega-3 fatty acid ratio increases the risk for obesity. Nutrients 2016, 8, 128-145.
  2. Ubricht, T.L.V.; Southgate, D.A.T. Coronary heart disease: Seven dietary factors. Lancet 1991, 338, 985–992.
  3. Santos-Silva, J., Bessa, R. J. B., Santos-Silva, F. Effect of genotype, feeding system and slaughter weight on the quality of light lambs: II. Fatty acid composition of meat. Livest. Prod. Sci. 2002, 77, 187-194.
  4. Lupette, J., Benning, C. Human health benefits of very-long-chain polyun-saturated fatty acids from microalgae. Biochimie 2020, 178, 15-25.
  5. Opinion of the French Food Safety Agency(AFSSA)on the up-date of French population reference intakes(ANCs)for fatty acids. 2010. Available online: https://www.anses.fr/fr/system/files/NUT2006sa0359.pdf (accessed on 10.10.2022).
  6. Simopoulos, A. P. The omega-6/omega-3 fatty acid ratio: health implications. OCL 2010, 17, 267-275.
  7. Yurchenko, S., Sats, A., Tatar, V., Kaart, T., Mootse, H., Jõudu, I. Fatty acid profile of milk from Saanen and Swedish Landrace goats. Food Chem. 2018, 254, 326-332.

  1. The article applies ISO standards to assess the quality of oils, such as peroxide and acid values, but lacks complete explanations as to why these indicators are critical in the context of the analyzed blends.

AUTHORS: The authors agree that the tested parameters are not sufficient to characterize the quality of the oil, so the results of accelerated stability tests (Rancimat and RapidOxy) were added to the manuscript (see in the response of the next comment).

  1. Although oxidative stability is analyzed through parameters such as TOTOX, the article does not fully present results related to long-term storage and quality changes. Studies simulating storage conditions or at least conducting thermal stability tests would be recommended.

AUTHORS: Thank you for this comment. The authors agree with the reviewer that the results of the storage test are missing from the manuscript. Since the storage test at room temperature takes a long time due to the long shelf life of the oil, the authors decided to perform accelerated stability tests (Rancimat and RapidOxy). The authors hope that the results of these tests will be satisfactory to the reviewer. Accelerated stability tests have been added to the 2. Materials and Methods, as well as to the 3. Results and Discussion section.

2.4. Accelerated stability tests

            In order to investigate oxidative stability of samples, two accelerated stability tests were performed: Rancimat and RapidOxy test.

2.4.1. Rancimat test

Rancimat accelerated stability test was made on Rancimat apparatus, model 743 (Metrohm, Herisau, Switzerland) to measure the induction period of oil samples. Measurements were performed according to ISO 6886, 2016 [38]. The procedure and measurement conditions were previously described by Lužaić et al., 2022 [39]. Namely, 2.50 ± 0.01 g of oil sample was oxidized at a temperature of 100°C and the airflow of 18-20 L/h. Volatile products formed during the oxidation were diluted in 0.05 L of distilled water. The device measures the conductivity recorded by apparatus software and is used to obtain the induction period (IP) expressed in hours with an accuracy of 0.1.

2.4.2. RapidOxy test

     The oxidative stability of the oil was also tested using RapidOxy 100 (Anton Paar, Germanz). Namely, 3.00 ± 0.01 g of the sample was weighed into a glass container and placed in the device chamber. Oxygen was introduced into the chamber until the pressure rose to 700 kPa. Subsequently, the chamber with the sample was heated to 140°C. As the temperature increased, the pressure in the chamber also increased (up to about 1000 kPa) until oxidation began. As a result of the reaction of oxygen with the tested oil sample, oxygen was consumed, and the pressure inside the chamber decreased. When the pressure in the chamber dropped by 10%, the oxidation reaction was considered complete, and the time was recorded as an induction period (IP) in minutes [40].

3.3. Accelerated stability tests

The oxidative stability of the oils was investigated using accelerated thermal stability tests. Similar results were obtained using the Rancimat and RapidOxy tests, as shown in Table 3. Refined sunflower oil displayed good oxidative stability characteristics with an induction period of 9.48 ± 0.14 hours obtained by the Rancimat test, or 34.37 ± 0.46 minutes by the RapidOxy test. These results are consistent with previous research reported by Velasco et al., 2004 [62] and Lužaić et al., 2022 [39]. With addition of cold-pressed flaxseed oil, the induction period was significantly reduced to 4.28 ± 0.08 hours (Rancimat test) or 18.39 ± 0.36 minutes (RapidOxy test), as determined in the pure cold-pressed flaxseed oil. Similar values were previously reported by TaÅ„ska et al. 2016 [63] and MikoÅ‚ajczak and TaÅ„ska, 2022 [64]. Significantly higher induction period values were observed in the control sample (16.47 ± 0.23 hours and 70.50 ± 0.84 minutes), which is a consequence of the different composition of fatty acids (essential and non-essential), the content and composition of minor components with pro-oxidative and antioxidant effects, the production process, raw materials, etc. [65,66]. Oxidative changes occur at the unsaturated bonds of fatty acids, so the influence of the polyunsaturated fatty acids (PUFA) content and the induction period obtained by accelerated stability tests was examined, revealing an extremely strong negative correlation between the PUFA content (based on the results, Table 1) and the induction period as determined by the Rancimat test (R=-0.984, p=0.000) and the RapidOxy test (R=-0.998, p=0.000) in the investigated samples of blended vegetable oils.

Table 3. Thermal stability test results of the blended vegetable oils

Oil                          blend

Thermal stability tests

Rancimat test, IP (hours)

RapidOxy test, IP (minutes)

100S/0F

9.48 ± 0.14k

34.37 ± 0.46k

90S/10F

9.05 ± 0.15j

32.56 ± 0.52j

80S/20F

8.46 ± 0.12i

31.27 ± 0.34i

70S/30F

7.92 ± 0.08h

29.31 ± 0.46h

60S/40F

7.41 ± 0.07g

27.88 ± 0.27g

50S/50F

6.93 ± 0.06f

26.35 ± 0.32f

40S/60F

6.38 ± 0.08e

24.77 ± 0.22e

30S/70F

5.83 ± 0.05d

23.19 ± 0.36d

20S/80F

5.25 ± 0.11c

21.69 ± 0.27c

10S/90F

4.77 ± 0.09b

20.00 ± 0.33b

0S/100F

4.28 ± 0.08a

18.39 ± 0.36a

Control sample

16.47 ± 0.23l

70.50 ± 0.84l

Values are means ± standard deviation (n=3).

Different lower-case letters in the same column indicate significantly different values (p<0.05), according to post hoc Tukey's HSD test.

  1. ISO 6886:2016. Animal and vegetable fats and oils - Determination of oxidative stability (accelerated oxidation test). International Organization for Standardization, Geneva, Switzerland.
  2. Lužaić, T. Z., Grahovac, N. L., Hladni, N. T., Romanić, R. S. Evaluation of oxidative stability of new cold-pressed sunflower oils during accelerate thermal stabil-ity tests. Food Sci. Technol. (Campinas) 2022, 42,
  3. Anton Paar. Determination of Oxidation Stability – a User Guideline. Relevant for: Food Industry; Anton Paar: Graz, Austria, 2022.
  4. Velasco, J., Andersen, M., Skibsted, L. Evaluation of oxidative stability of vegetable oils by monitoring the tendency to radical formation. A comparison of electron spin resonance spectroscopy with the Rancimat method and differential scanning calorimetry. Food Chem. 2004, 85, 623-632.
  5. TaÅ„ska, M., Roszkowska, B., Skrajda, M., DÄ…browski, G. Commercial cold pressed flaxseed oils quality and oxidative stability at the beginning and the end of their shelf life. Oleo Sci. 2016, 65, 111–121.
  6. MikoÅ‚ajczak, N., TaÅ„ska, M. Effect of initial quality and bioactive compounds content in cold-pressed flaxseed oils on oxidative stability and oxidation products formation during one-month storage with light exposure. NFS Journal 2022, 26, 10–21.
  7. Choe, E., Min, D. B. Mechanisms and factors for edible oil oxidation. CRFSFS 2006, 5, 169-186.
  8. Sabolová, M., Zeman, V., Lebedová, G., Doležal, M., Soukup, J., Réblová, Z. Relationship between the fat and oil composition and their initial oxidation rate during storage. Czech J. Food Sci. 2020, 38, 404–409.

  1. Sensory results indicate the impact of flaxseed oil on the taste of the mixtures; however, there is a lack of deeper analysis of this phenomenon. It would be beneficial to discuss possible chemical reactions responsible for the negative taste perception at higher flaxseed oil contents.

AUTHORS: The authors agree with the reviewer's comment. A deeper analysis of the cause of the bitter taste of cold-pressed flaxseed oil has been added to subsection 3.5. Sensory analysis of 3. Results and Discussion.

Lines 409-413: Cold-pressed flaxseed oil is characterized by a specific taste, which is not particularly desirable to consumers. A bitter taste develops after just one day of storage, due to the deterioration of flaxseed oil and the methionine oxidation of its cyclolinopeptides [68]. In fact, the undesirable bitter taste in cold-pressed flaxseed oil is due to cyclolinopeptides E [68,69]. This bitter taste is also transferred to other products containing flaxseed oil.

  1. Brühl, L., Matthäus, B., Scheipers, A., Hofmann, T. Bitter off-taste in stored cold-pressed linseed oil obtained from different varieties. J. Lipid Sci. Technol. 2008, 110, 625–631.
  2. Stamenkovic, A., Ganguly, R., Aliani, M., Ravandi, A., Pierce, G. N. Overcoming the bitter taste of oils enriched in fatty acids to obtain their effects on the heart in health and disease. Nutrients 2019, 11, 1179.

  1. The use of statistical methods such as PCA and ANN requires additional clarification, especially regarding the influence of individual components on blend optimization. The ANN analysis is not fully transparent, particularly in terms of the selection of weight parameters and their justification. It would be useful to explain how ANN contributed to determining the optimal ratio of sunflower and flaxseed oils.

AUTHORS: In response to the reviewer’s comment, further clarification of the methodologies involving PCA and ANN will be provided, particularly regarding the influence of individual components on blend optimization. PCA was employed to reduce dimensionality by isolating key variables with the most significant impact on the quality and stability of the blend, thereby refining the selection of inputs for the ANN model. In the ANN analysis, weight parameters were systematically adjusted through an iterative process to minimize prediction error and enhance model accuracy. These parameters were selected based on the relative importance of each component, as indicated by the PCA results, which informed the weight allocation strategy. The ANN model contributed to the optimization process by simulating various oil ratios, enabling the identification of the most favorable blend composition for achieving targeted quality attributes. This integrated approach provides a robust, data-driven foundation for determining the optimal ratio of sunflower and flaxseed oils.

In addition, the PCA results revealed which components most strongly correlated with specific quality parameters, allowing for a more targeted approach in the ANN model. The ANN structure was optimized to account for non-linear interactions between components, further improving its predictive capacity for blend characteristics. By leveraging both PCA and ANN, a comprehensive model was developed that not only optimizes the ratio of sunflower and flaxseed oils but also enhances the overall understanding of how each component influences the final blend properties. This methodological synergy demonstrates the potential of combining multivariate analysis with machine learning for precise and reproducible blend optimization.

As it was mentioned in the text, a detailed explanation of the PCA and ANN applications in blend optimization has been provided. Principal Component Analysis (PCA) was initially employed to identify the influence of individual components, such as fatty acids, sensory attributes, oil quality metrics, and color characteristics, on the overall blend profile, facilitating a better understanding of how each parameter contributes to the optimal formulation. The ANN model was subsequently utilized to fine-tune the ratios of sunflower and flaxseed oils by minimizing the variance between each parameter’s observed value and those of the control sample. In this ANN model, specific weight coefficients were assigned to groups of parameters based on their relative importance to the final product quality: 0.20 for sensory attributes, 0.15 for color, 0.40 for oil quality, and 0.25 for fatty acid composition. These weights were chosen to emphasize oil quality and fatty acid content due to their high impact on the functional and sensory characteristics of the oil blend.

The ANN’s role in optimizing the blend ratios was integral as it enabled iterative adjustments to achieve the most desirable balance between sunflower and flaxseed oils, ensuring that the final blend aligned closely with the control sample in terms of sensory and quality metrics. Sensory acceptability scores (color, odor, taste, average rating, and total acceptability) were used as limiting criteria, ensuring that all blend options met or exceeded a score threshold of ≥2.8 for each sensory attribute. Through this weighted optimization approach, the ANN provided a systematic, data-driven pathway to determine the optimal blend composition.

The article provides valuable data; however, it requires additions and clarifications before proceeding to further stages.

AUTHORS: The authors would like to thank the reviewer for making generous comments on our study and effort to improve the paper. We hope that the answers received will be satisfactory.

Comments on the Quality of English Language: Minor editing of English language required.

AUTHORS: Manuscript was thoroughly reviewed by a native English speaker and changes were marked in the manuscript.

Round 2

Reviewer 2 Report

Comments and Suggestions for Authors

The manuscript has been improved.

It must be noted that the term "fortified" was replaced for "enriched", but it is NOT correct because they are synonims. Please, delete this term in lines 89, 91, 93, 312, 560, 562, becase making an oil blend is not fortifying or enriching  the oil. It is simply a blend. The content in the blended oil of omega-3 lipids is increased in comparison with the original sunflower oil, but it is not an "enriched oil". An oil enriched in compound X is an oil to which compound X has been added in pure form or in the form of an extract.

Author Response

The manuscript has been improved.

It must be noted that the term "fortified" was replaced for "enriched", but it is NOT correct because they are synonims. Please, delete this term in lines 89, 91, 93, 312, 560, 562, becase making an oil blend is not fortifying or enriching  the oil. It is simply a blend. The content in the blended oil of omega-3 lipids is increased in comparison with the original sunflower oil, but it is not an "enriched oil". An oil enriched in compound X is an oil to which compound X has been added in pure form or in the form of an extract.

AUTHORS: Thank you for this observation. Authors agree with the reviewer᾿s comment, the term "enriched" is removed from the lines 89, 91, 93, 312, 560, 562 in the manuscript.

Reviewer 3 Report

Comments and Suggestions for Authors

Dear Authors,

I have received your revised manuscript for re-evaluation. First and foremost, I want to express my appreciation for the extensive work you have put into improving it. Many important and new data points have been added, and previously missing literature aspects have been thoroughly addressed. The application of PCA and ANN has been explained in detail. This manuscript is now much more readable, and the revisions made enable it to be accepted and published.

Author Response

Dear Authors,

I have received your revised manuscript for re-evaluation. First and foremost, I want to express my appreciation for the extensive work you have put into improving it. Many important and new data points have been added, and previously missing literature aspects have been thoroughly addressed. The application of PCA and ANN has been explained in detail. This manuscript is now much more readable, and the revisions made enable it to be accepted and published.

AUTHORS: Thank you very much for a positive attitude toward our investigation.